# Separable Physics-Informed Neural Networks

**Junwoo Cho**[1*]   **Seungtae Nam**[1*]   **Hyunmo Yang**[1]
**Seok-Bae Yun**[2]   **Youngjoon Hong**[3]   **Eunbyung Park**[1,4†]

[1]Department of Artificial Intelligence, Sungkyunkwan University
[2]Department of Mathematics, Sungkyunkwan University
[3]Department of Mathematical Sciences, KAIST
[4]Department of Electrical and Computer Engineering, Sungkyunkwan University

## Abstract

Physics-informed neural networks (PINNs) have recently emerged as promising data-driven PDE solvers showing encouraging results on various PDEs. However, there is a fundamental limitation of training PINNs to solve multi-dimensional PDEs and approximate highly complex solution functions. The number of training points (collocation points) required on these challenging PDEs grows substantially, but it is severely limited due to the expensive computational costs and heavy memory overhead. To overcome this issue, we propose a network architecture and training algorithm for PINNs. The proposed method, *separable PINN (SPINN)*, operates on a per-axis basis to significantly reduce the number of network propagations in multi-dimensional PDEs unlike point-wise processing in conventional PINNs. We also propose using forward-mode automatic differentiation to reduce the computational cost of computing PDE residuals, enabling a large number of collocation points ($> 10^7$) on a single commodity GPU. The experimental results show drastically reduced computational costs ($62\times$ in wall-clock time, $1,394\times$ in FLOPs given the same number of collocation points) in multi-dimensional PDEs while achieving better accuracy. Furthermore, we present that SPINN can solve a chaotic (2+1)-d Navier-Stokes equation significantly faster than the best-performing prior method (9 minutes vs 10 hours in a single GPU), maintaining accuracy. Finally, we showcase that SPINN can accurately obtain the solution of a highly nonlinear and multi-dimensional PDE, a (3+1)-d Navier-Stokes equation. For visualized results and code, please see `https://jwcho5576.github.io/spinn.github.io/`.

## 1   Introduction

Solving partial differential equations (PDEs) has been a long-standing problem in various science and engineering domains. Finding analytic solutions requires in-depth expertise and is often infeasible in many useful and important PDEs [10]. Hence, numerical approximation methods to solutions have been extensively studied [17], e.g., spectral methods [3], finite volume methods (FVM) [9], finite difference method (FDM) [39], and finite element methods (FEM) [47]. While successful, classical methods have several limitations, such as expensive computational costs, requiring sophisticated techniques to support multi-physics and multi-scale systems, and the curse of dimensionality in high dimensional PDEs.

With the vast increases in computational power and methodological advances in machine learning, researchers have explored data-driven and learning-based methods [4, 24, 30, 38]. Among the promising methods, physics-informed neural networks (PINNs) have recently emerged as new data-driven PDE solvers for both forward and inverse problems [34]. PINNs employ neural networks and

---

[*]Equal contribution.
[†]Corresponding author.

37th Conference on Neural Information Processing Systems (NeurIPS 2023).

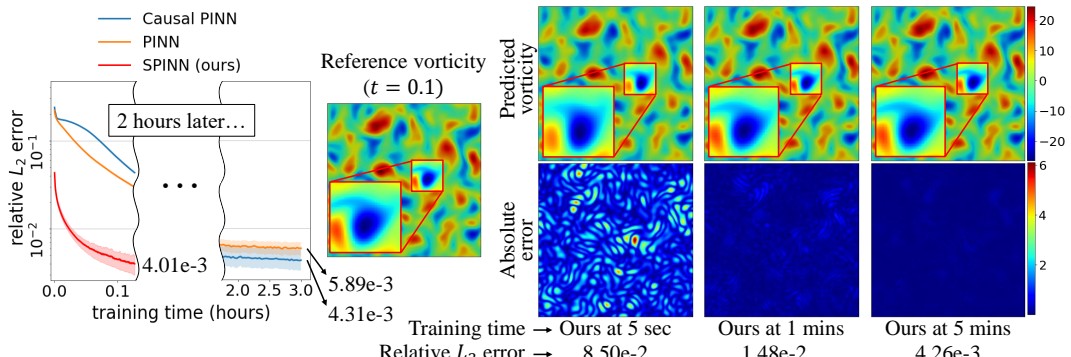

Figure 1: Training speed (w/ a single GPU) of our model compared to the causal PINN [43] in (2+1)-d Navier-Stokes equation of time interval [0, 0.1].

gradient-based optimization algorithms to represent and obtain the solutions, leveraging automatic differentiation to enforce the physical constraints of underlying PDE. It has enjoyed great success in various forward and inverse problems thanks to its numerous benefits, such as flexibility in handling a wide range of forward and inverse problems, mesh-free solutions, and not requiring observational data, hence, unsupervised training.

Despite its advantages and promising results, there is a fundamental limitation of training PINN to solve multi-dimensional PDEs and approximate very complex solution functions. It primarily stems from using coordinate-based MLP architectures to represent the solution function, which takes input coordinates and outputs corresponding solution quantities. For each training point, computing PDE residual loss involves multiple forward and backward propagations, and the number of training points (collocation points) required to solve multi-dimensional PDEs and obtain more accurate solutions grows substantially. The situation deteriorates as the dimensionality of the PDE or the solution's complexity increases.

Recent studies have presented empirical evidence showing that choosing a larger batch size (i.e., a large number of collocation points) in training PINNs leads to enhanced precision [34, 36, 37, 43]. Furthermore, more collocation points also accelerate the convergence speed due to the scaling characteristic between the batch size and the learning rate (the larger the batch size, the higher the learning rate) [12, 27, 40]. As long as the computational resources allow, we have the flexibility to utilize a substantial number of training points in PINNs since they can be continuously sampled from the input domain in an unsupervised training manner.

We propose a novel PINN architecture, *separable PINN (SPINN)*, which utilizes forward-mode* automatic differentiation (AD) to enable a large number of collocation points ($> 10^7$ in a single GPU), reducing the computational cost of solving multi-dimensional PDEs. Instead of feeding every multi-dimensional coordinate into a single MLP, we use separated sub-networks, in which each sub-network takes independent one-dimensional coordinates as input. The final output is generated by an aggregation module such as simple outer product and element-wise summation where the predicted solution can be interpreted by low-rank tensor approximation [25] (Fig. 4b). The suggested architecture obviates the need to query every multi-dimensional coordinate input pair, exponentially reducing the number of network propagations to generate a solution, $\mathcal{O}(N^d) \rightarrow \mathcal{O}(Nd)$, where $N$ is the resolution of the solution for each dimension, and $d$ is the dimension of the system.

We have conducted comprehensive experiments on the representative PDEs to show the effectiveness of the suggested method. The experimental results demonstrate that the training runtime of the conventional PINN increase linearly with the number of collocation points, while the proposed model shows logarithmic growths. This allows SPINN to accommodate orders of magnitude larger number of collocation points *in a single batch* during training. We also show that given the same number of training points, SPINN improves wall-clock training time up to by $62\times$ on commodity GPUs and FLOPs up to by $1,394\times$ while achieving better accuracy. Furthermore, with large-scale collocation points, SPINN can solve a turbulent Navier-Stokes equation much faster than the state-of-the-art PINN method [43] (9 minutes vs 10 hours in a single GPU) without bells and whistles, such as causal

---

*a.k.a. forward accumulation mode or tangent linear mode.

inductive bias in the loss function (Fig. 1). Our experimental results of the Navier-Stokes equation show that SPINN can solve highly nonlinear PDEs and is sufficiently expressive to represent complex functions. This is further supported by the provided theoretical result, potentiating the use of our method for more challenging and various PDEs.

## 2 Related Works

**Physics-informed neural networks.** Physics-Informed Neural Networks (PINNs) [34] have received great attention as a promising learning-based PDE solver. Given the underlying PDE and initial, boundary conditions embedded in a loss function, a coordinate-based neural network is trained to approximate the desired solution. Since its inception, many techniques have been studied to improve training PINNs for more challenging PDE systems [26, 43, 44], or to accelerate the training speed [20, 37]. Our method is orthogonal to most of the previously suggested techniques above and improves PINNs' training from a computational perspective.

**The effect of collocation points.** The residual loss of PINNs is calculated by Monte Carlo integration, not an exact definite integral. Therefore, it inherits a core property of Monte Carlo methods [15]: the impact of the number of sampled points. The importance of sampling strategy and the number of collocation points in training PINNs has been highlighted by recent works [7, 23, 36, 43]. Especially, Sankaran et al. [36] empirically found that training with a large number of collocation points is unconditionally favorable for PINNs in terms of accuracy and convergence speed. Another line of research established a theoretical upper bound of PINN's statistical error with respect to the number of collocation points [23]. It showed that a larger number of collocation points are required to use a bigger network size for training. Our work builds on this evidence to bring PINNs out in more practical scenarios and overcome their limitation.

**Derivative computations in scientific machine learning.** Embedding physical constraints in the loss function is widely used in scientific machine learning, and computing the derivatives is an essential process for this formulation. Several works employed numerical differentiation [33, 35, 42], or hybrid approach [5, 37, 46] with AD for the calculation, since numerical methods such as finite differences do not need to back-propagate through the network. However, they are still burdened by a computational complexity of $\mathcal{O}(N^d)$, thereby limiting them to handle large-scale collocation points or meshes. Furthermore, numerical differentiation has truncation errors depending on the step size. Employing Taylor-mode AD [13] in training PINNs was introduced by causal PINN [43] to handle high-order PDEs such as Kuramoto–Sivashinsky equation [28]. To the best of our knowledge, the proposed method is the first approach to leverage forward-mode AD in training PINNs, which is fully applicable to both time-dependent and independent PDEs and does not incur any truncation errors.

**Multiple MLP networks.** Employing multiple MLPs for PINNs has been introduced by several works to utilize parallelized training [19, 21, 32]. They share the same concept of dividing the entire spatio-temporal domain and training multiple individual MLPs on each sub-domain. Although these methods showed promising results, they still suffer from the fundamental problem of heavy computation as the number of collocation points increases. While these methods decompose input domains, and each small MLP is used to cover a particular sub-domain, we decompose input dimensions and solve PDEs over the entire domain cooperated by all separated MLPs. In terms of the model architecture and function representation, our work is also related to NAM [1], Haghighat et al. [14], and CoordX [31]. NAM [1] suggested separated network architectures and inputs, but only for achieving the interpretability of the model's prediction in multi-task learning. Haghighat et al. [14] used multiple MLPs to individually predict each component of the *output* vector. CoordX's [31] primary purpose is to reconstruct natural signals, such as images or 3D shapes. Hence, they are not motivated to improve the efficiency of computing higher-order gradients. In addition, they had to use additional layers after the feature merging step, which made their model more computationally expensive than ours. Furthermore, in neural fields [45], there is an explicit limitation in the number of ground truth data points (e.g., the number of pixels in an image). Therefore, CoordX cannot fully maximize the advantage of enabling a large number of input coordinates. We focus on solving PDEs and carefully devise the architecture to exploit forward-mode AD to efficiently compute PDE residual losses.

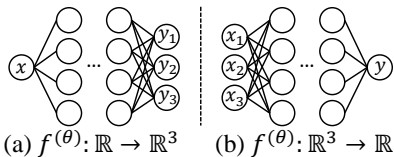

(a) $f^{(\theta)}: \mathbb{R} \to \mathbb{R}^3$    (b) $f^{(\theta)}: \mathbb{R}^3 \to \mathbb{R}$

Figure 2: Simple neural networks with different input and output dimensions. To compute $\frac{\partial y}{\partial x}$, (a) requires one forward pass using forward-mode AD or three backward passes using reverse-mode AD, (b) requires three forward passes using forward-mode AD or one backward pass using reverse-mode AD.

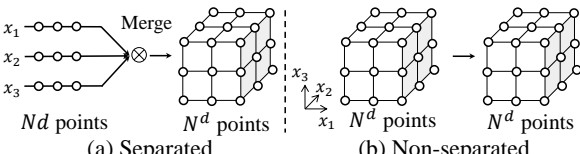

$Nd$ points    $N^d$ points    $N^d$ points    $N^d$ points

(a) Separated      (b) Non-separated

Figure 3: An illustrative example of separated approach vs non-separated approach when $f^{(\theta)}: \mathbb{R}^d \to \mathbb{R}$ (an example case of $N = 3$, $d = 3$ is shown above). The number of JVP (forward-mode) evaluations (propagations) of computing the Jacobian for separated approach (a) is $Nd$, while the number of VJP (reverse-mode) evaluations for non-separated approach (b) is $N^d$.

## 3 Preliminaries: Forward/Reverse-mode AD

For the completeness of this paper, we start by briefly introducing the two types of AD and how Jacobian matrices are evaluated. For clarity, we will follow the notations used in [2] and [13]. Suppose our function $f : \mathbb{R}^n \to \mathbb{R}^m$ is a two-layer MLP with 'tanh' activation. The left-hand side of Tab. 1 demonstrates a single forward trace of $f$. To obtain a $m \times n$ Jacobian matrix $\mathbb{J}_f$ in forward-mode, we compute the Jacobian-vector product (JVP),

Table 1: An example of forward and reverse-mode AD in a two-layers tanh MLP. Here $v_0$ denotes the input variable, $v_k$ the primals, $\dot{v}_k$ the tangents, $\bar{v}_k$ the adjoints, and $W_1, W_2$ the weight matrices. Biases are omitted for brevity.

| Forward primal trace | Forward tangent trace | Backward adjoint trace |
|---|---|---|
| $v_0 = x$ | $\dot{v}_0 = \dot{x}$ | $\bar{x} = \bar{v}_0$ |
| $v_1 = W_1 \cdot v_0$ | $\dot{v}_1 = W_1 \cdot \dot{v}_0$ | $\bar{v}_0 = \bar{v}_1 \cdot \frac{\partial v_1}{\partial v_0} = \bar{v}_1 \cdot W_1$ |
| $v_2 = \tanh(v_1)$ | $\dot{v}_2 = \tanh'(v_1) \circ \dot{v}_1$ | $\bar{v}_1 = \bar{v}_2 \cdot \frac{\partial v_2}{\partial v_1} = \bar{v}_2 \circ \tanh'(v_1)$ |
| $v_3 = W_2 \cdot v_2$ | $\dot{v}_3 = W_2 \cdot \dot{v}_2$ | $\bar{v}_2 = \bar{v}_3 \cdot \frac{\partial v_3}{\partial v_2} = \bar{v}_3 \cdot W_2$ |
| $y = v_3$ | $\dot{y} = \dot{v}_3$ | $\bar{v}_3 = \bar{y}$ |

$$\mathbb{J}_f r = \begin{bmatrix} \frac{\partial y_1}{\partial x_1} & \cdots & \frac{\partial y_1}{\partial x_n} \\ \vdots & \ddots & \vdots \\ \frac{\partial y_m}{\partial x_1} & \cdots & \frac{\partial y_m}{\partial x_n} \end{bmatrix} \begin{bmatrix} \frac{\partial x_1}{\partial x_i} \\ \vdots \\ \frac{\partial x_n}{\partial x_i} \end{bmatrix}, \tag{1}$$

for $i \in \{1, \ldots, n\}$. The forward-mode AD is a one-phase process: while tracing primals (intermediate values) $v_k$, it continues to evaluate and accumulate their tangents $\dot{v}_k = \partial v_k / \partial x_i$ (the middle column of Tab. 1). This is equivalent to decomposing one large JVP into a series of JVPs by the chain rule and computing them from right to left. A run of JVP with the initial tangents $\dot{v}_0$ as the first column vector of an identity matrix $I_n$ gives the first column of $\mathbb{J}_f$. Thus, the full Jacobian can be obtained in $n$ forward passes.

On the other hand, the reverse-mode AD computes vector-Jacobian product (VJP):

$$r^\top \mathbb{J}_f = \begin{bmatrix} \frac{\partial y_j}{\partial y_1} & \cdots & \frac{\partial y_j}{\partial y_m} \end{bmatrix} \begin{bmatrix} \frac{\partial y_1}{\partial x_1} & \cdots & \frac{\partial y_1}{\partial x_n} \\ \vdots & \ddots & \vdots \\ \frac{\partial y_m}{\partial x_1} & \cdots & \frac{\partial y_m}{\partial x_n} \end{bmatrix}, \tag{2}$$

for $j \in \{1, \ldots, m\}$, which is the reverse-order operation of JVP. This is a two-phase process. The first phase corresponds to forward propagation, storing all the primals, $v_k$, and recording the elementary operations in the computational graph. In the second phase, the derivatives are computed by accumulating the adjoints $\bar{v}_k = \partial y_j / \partial v_k$ (the right-hand side of Tab. 1). Since VJP builds one row of a Jacobian at a time, it takes $m$ evaluations to obtain the full Jacobian. To sum up, the forward-mode is more efficient for a tall Jacobian ($m > n$), while the reverse-mode is better suited for a wide Jacobian ($n > m$). Fig. 2 shows an illustrative example and please refer to Baydin et al. [2] for more details.

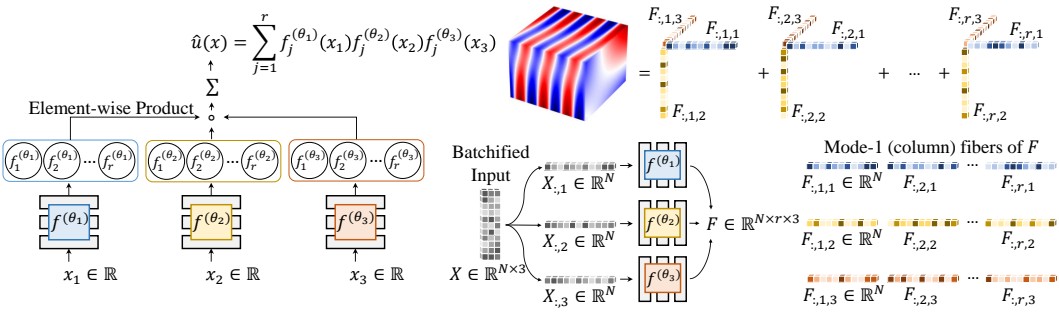

| (a) Architecture (single input) | (b) Low-rank tensor decomposition (batchified input) |

Figure 4: (a) SPINN architecture in a 3-dimensional system. To solve a $d$-dimensional PDE, our model requires $d$ body MLP networks, each of which takes individual scalar coordinate values as input and gives $r$-dimensional feature vector. The final output is obtained by element-wise product and summation. (b) Construction process of the entire discretized solution tensor when the inputs are given in batches. Each outer product between the column vectors $F_{:,j,i}$ from the feature tensor $F$ constructs a rank-1 tensor and summing all the $r$ tensors gives a rank-$r$ tensor. The output tensor of SPINN can be interpreted as a low-rank decomposed representation of a solution.

## 4 Separable PINN

### 4.1 Forward-Mode AD with Separated Functions

We demonstrate that leveraging forward-mode AD and separating the function into multiple functions with respect to input axes can significantly reduce the cost of computing Jacobian. In the proposed separated approach (Fig. 3a), we first sample $N$ one-dimensional coordinates on each of $d$ axes, which makes a total of $Nd$ batch size. Next, these coordinates are fed into $d$ individual functions. Let $f$ be a function which takes coordinate as input to produce feature representation and we denote the number of operations of $f$ as $\mathsf{ops}(f)$. Then, a feature merging function $h$ is used to construct the solution of the entire $N^d$ discretized points. The amount of computations for AD is known to be 2~3 times more expensive than the forward propagation [2, 13]. According to Fig. 3 and with the scale constants $c_f, c_h \in [2, 3]$ we can approximate the total number of operations to compute the Jacobian matrix of the proposed separated approach (Fig. 3a).

$$\mathcal{C}_{\text{sep}} = N d c_f \mathsf{ops}(f) + N^d c_h \mathsf{ops}(h). \tag{3}$$

For a non-separated approach (Fig. 3b),

$$\mathcal{C}_{\text{non-sep}} = N^d c_f \mathsf{ops}(f), \tag{4}$$

If we can make $\mathsf{ops}(h)$ sufficiently small, then the ratio $\mathcal{C}_{\text{sep}}/\mathcal{C}_{\text{non-sep}}$ becomes $\frac{Nd}{N^d} \ll 1$, quickly converging to 0 as $d$ and $N$ increases. Our separated approach has linear complexity with respect to $N$ in network propagations, implying it can obtain more accurate solutions (high-resolution) efficiently.

### 4.2 Network Architecture

Fig. 4a illustrates the overall SPINN architecture, parameterizing multiple separated functions with neural networks. SPINN consists of $d$ body-networks (MLPs), each of which takes an individual 1-dimensional coordinate component as an input. Each body-network $f^{(\theta_i)} : \mathbb{R} \to \mathbb{R}^r$ (parameterized by $\theta_i$) is a vector-valued function which transforms the coordinates of $i$-th axis into a $r$-dimensional feature representation. The final prediction is computed by feature merging:

$$\hat{u}(x_1, x_2, \ldots, x_d) = \sum_{j=1}^{r} \prod_{i=1}^{d} f_j^{(\theta_i)}(x_i) \tag{5}$$

where $\hat{u} : \mathbb{R}^d \to \mathbb{R}$ is the predicted solution function, $x_i \in \mathbb{R}$ is a coordinate of $i$-th axis, and $f_j^{(\theta_i)}$ denotes the $j$-th element of $f^{(\theta_i)}$. We used 'tanh' activation function throughout the paper. As shown in Eq. 5, the feature merging operation is a simple product ($\Pi$) and summation ($\Sigma$) which

corresponds to the merging function $h$ described in Eq. 3. Due to its simplicity, $h$ operations are much cheaper than operations in MLP layers (i.e., $\mathsf{ops}(h) \ll \mathsf{ops}(f)$ in Eq. 3). Note that SPINN can also approximate any $m$-dimensional vector functions $\hat{u} : \mathbb{R}^d \to \mathbb{R}^m$ by using a larger output feature size (see section D.4 in the appendix for details).

The collocation points of our model and conventional PINNs have a distinct difference (Fig. 5). Both are *uniformly* evaluated on a $d$-dimensional hypercube, but collocation points of SPINN form a lattice-like structure, which we call as *factorizable coordinates*. In SPINN, 1-dimensional input points from each axis are randomly sampled, and $d$-dimensional points are generated via the cartesian product of the point sets from each axis. On the other hand, non-factorizable coordinates are randomly sampled points without any structure. Factorizable coordinates with our separated MLP architecture enable us to evaluate functions on dense ($N^d$) collocation points with a small number ($Nd$) of input points.



(a) Factorizable coordinates (b) Non-factorizable coordinates

Figure 5: An illustrative 2-dimensional example of (a) factorizable and (b) non-factorizable coordinates. (a) has a lattice-like structure, where SPINN can be evaluated on more dense collocation points with fewer input points. Conventional PINNs sample non-factorizable coordinates which do not have any structures.

In practice, the input coordinates are given in a batch during training and inference. Assume that $N$ input coordinates (training points) are sampled from each axis. Note that the sampling resolutions for each axis need not be the same. The input coordinates $X \in \mathbb{R}^{N \times d}$ is now a matrix. The batchified form of feature representation $F \in \mathbb{R}^{N \times r \times d}$ and Eq. 5 now becomes

$$\hat{U}(X_{:,1}, X_{:,2}, \ldots, X_{:,d}) = \sum_{j=1}^{r} \bigotimes_{i=1}^{d} F_{:,j,i}, \tag{6}$$

where $\hat{U} \in \mathbb{R}^{N \times N \times \cdots \times N}$ is the discretized solution tensor, $\bigotimes$ denotes outer product, $F_{:,:,i} \in \mathbb{R}^{N \times r}$ is an $i$-th frontal slice matrix of tensor $F$, and $F_{:,j,i} \in \mathbb{R}^N$ is the $j$-th column of the matrix $F_{:,:,i}$. Fig. 4b shows an illustrative procedure of Eq. 6. Due to its structural input points and outer products between feature vectors, SPINN's solution approximation can be viewed as a low-rank tensor decomposition where the feature size $r$ is the rank of the reconstructed tensor. Among many decomposition methods, SPINN corresponds to CP-decomposition [16], which approximates a tensor by finite summation of rank-1 tensors. While traditional methods use iterative methods such as alternating least-squares (ALS) or alternating slicewise diagonalization (ASD) [22] to directly fit the decomposed vectors, we train neural networks to learn the decomposed vector representation and approximate the solution functions in continuous input domains. This, in turn, allows for the calculation of derivatives with respect to arbitrary input coordinates.

## 4.3 Gradient Computation of SPINN

In this section, we show that the number of JVP (forward-mode AD) evaluations for computing the full gradient of SPINN ($\nabla \hat{u}(x)$) is $Nd$, where $N$ is the number of coordinates sampled from each axis and $d$ is the input dimension. According to Eq. 5, the $i$-th element of $\nabla \hat{u}(x)$ is:

$$\frac{\partial \hat{u}}{\partial x_i} = \sum_{j=1}^{r} f_j^{(\theta_1)}(x_1) f_j^{(\theta_2)}(x_2) \ldots \frac{\partial f_j^{(\theta_i)}(x_i)}{\partial x_i} \ldots f_j^{(\theta_d)}(x_d). \tag{7}$$

Computing this derivative requires feature representations $f_j^{(\theta_i)}$, which can be reused from the forward pass results computed beforehand. The entire $r$ components of the feature derivatives $\partial f^{(\theta_i)}(x_i)/\partial x_i : \mathbb{R} \to \mathbb{R}^r$ can be obtained by a single pass thanks to forward-mode AD. To obtain the full gradient $\nabla \hat{u}(x) : \mathbb{R}^d \to \mathbb{R}^d$, we iterate the calculation of Eq. 7 over $d$ times, switching the input axis $i$. Also, since each iteration involves $N$ training samples, the total number of JVP evaluations becomes $Nd$, which is consistent with Eq. 3. Note that any $p$-th order derivative $\partial^p \hat{u}/\partial x_i^p$ can also be obtained in the same way.

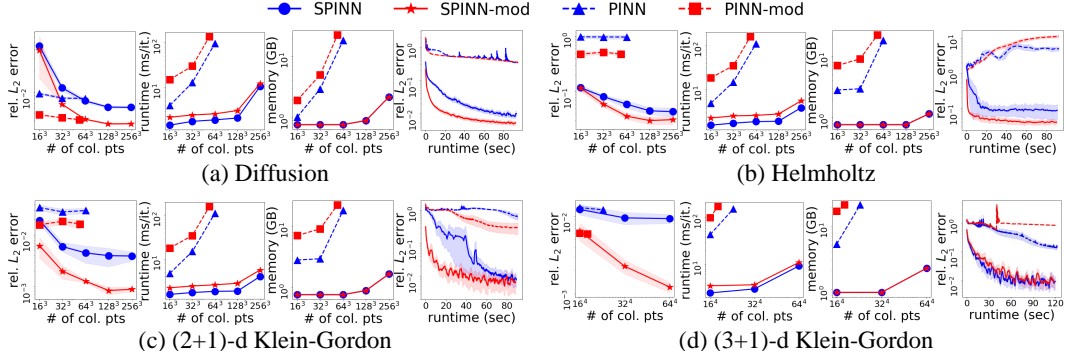

Figure 6: Overall results of (a) diffusion, (b) Helmholtz, (c) (2+1)-d Klein-Gordon, and (d) (3+1)-d Klein-Gordon experiments. It shows comparisons among PINN, PINN with modified MLP (PINN-mod), SPINN, and SPINN with modified MLP (SPINN-mod). For each experiment, the first to third columns show the relative error, runtime, and GPU memory versus different numbers of collocation points, respectively. The rightmost column shows the training curves of each model when the number of collocation points is $64^3$ ($54^3$ for PINN with modified MLP.) For (3+1)-d Klein-Gordon, the training curve is plotted when each model is trained with the number of collocation points of $16^4$. The error and the training curves are averaged over 7 different runs and 70% confidence intervals are provided. Note that the y-axis scale of every plot is a log scale.

## 4.4 Universal Approximation Property

It is widely known that neural networks with sufficiently many hidden units have the expressive power of approximating any continuous functions [6, 18]. However, it is not straightforward that our suggested architecture enjoys the same capability. For the completeness of the paper, we provide a universal approximation property of the proposed method.

**Theorem 1.** *(Proof in appendix) Let $X, Y$ be compact subsets of $\mathbb{R}^d$. Choose $u \in L^2(X \times Y)$. Then, for arbitrary $\varepsilon > 0$, we can find a sufficiently large $r > 0$ and neural networks $f_j$ and $g_j$ such that*

$$\left\| u - \sum_{j=1}^{r} f_j g_j \right\|_{L^2(X \times Y)} < \varepsilon. \tag{8}$$

By repeatedly applying Theorem 1, we can show that SPINN can approximate any functions in $L^2$ in the high-dimensional input space. An approximation property for a broader function space is a fruitful research area, and we leave it to future work. To support the expressive power of the suggested method, we empirically showed that SPINN can accurately approximate solution functions of various challenging PDEs, including diffusion, Helmholtz, Klein-Gordon, and Navier-Stokes equations.

# 5 Experiments

## 5.1 Experimental setups

We compared SPINN against vanilla PINN [34] on 3-d (diffusion, Helmholtz, Klein-Gordon, and Navier-Stokes equation) and 4-d (Klein-Gordon and Navier-Stokes) PDE systems. Every experiment was run on a different number of collocation points, and we also applied the modified MLP introduced in [44] to both PINN and SPINN. For 3-d systems, the number of collocation points of $64^3$ was the upper limit for the vanilla PINN ($54^3$ for modified MLP) when we trained with a single NVIDIA RTX3090 GPU with 24GB of memory. However, the memory usage of our model was significantly smaller, enabling SPINN to use a larger number of collocation points (up to $256^3$) to get more accurate solutions. This is because SPINN stores a much smaller batch of tensors, which are the primals ($v_k$ in Tab. 1) while building the computational graph. All reported error metrics are average relative $L_2$ errors computed by $\|\hat{u} - u\|^2 / \|u\|^2$, where $\hat{u}$ is the model prediction, and $u$ is the reference solution. Every experiment is performed seven times (three times for (2+1)-d and five times for (3+1)-d Navier-Stokes, respectively) with different random seeds. More detailed experimental settings are provided in the appendix.

## 5.2 Results

Fig. 6 shows the overall results of forward problems on three 3-d systems (diffusion, Helmholtz, and Klein-Gordon) and one 4-d system (Klein-Gordon). SPINN is significantly more computationally efficient than the baseline PINN in wall-clock run-time. For every PDE, SPINN with the modified MLP found the most accurate solution. Furthermore, when the number of collocation points grows exponentially, the memory usage and the actual run-time of SPINN increase almost linearly. We also confirmed that we can get more accurate solutions with more collocation points. This training characteristic of SPINN substantiates that our method is very effective for solving multi-dimensional PDEs. Furthermore, with the help of the method outlined in Griewank and Walther [13], we estimated the FLOPs for evaluating the derivatives. Compared to the baseline, SPINN requires $1,394\times$ fewer operations to compute the forward pass, first and second order derivatives. Further details regarding the FLOPs estimation are provided in section C of the appendix. In the following sections, we give more detailed descriptions and analyses of each experiment.

**Diffusion Equation**    When trained with $128^3$ collocation points, SPINN with the modified MLP finds the most accurate solution. Furthermore, when trained with the same number of collocation points ($64^3$), SPINN-mod is $52\times$ faster and $29\times$ more memory efficient than the baseline with modified MLP. You can find the visualized solution in the appendix. Since we construct the solution relatively simple compared to other PDE experiments, baseline PINNs also find a comparatively good solution. However, our model finds a more accurate solution with a larger number of collocation points with minor computational overhead. The exact numerical values are provided in the appendix.

**Helmholtz Equation**    The results of the Helmholtz experiments are shown in Fig. 6b. Due to stiffness in the gradient flow, conventional PINNs hinder finding accurate solutions. Therefore, a modified MLP and learning rate annealing algorithm is suggested to mitigate such phenomenon [44]. We found that even PINNs with modified MLP fail to solve the equation when the solution is complex (contains high-frequency components). However, SPINN obtains one order of magnitude lower relative error without bells and whistles. This is because each body network of the SPINN learns an individual 1-dimensional function which mitigates the burden of representing the entire 3-dimensional function. This makes our model much easier to learn complex functions. Given the same number of collocation points ($64^3$), the training speed of our proposed model with modified MLP is $62\times$ faster, and the memory usage is $29\times$ smaller than the baseline with modified MLP. The exact numerical values are provided in the appendix.

**Klein-Gordon Equation**    Fig. 6c shows the results. Again, our method shows the best performance in terms of accuracy, runtime ($62\times$), and memory usage ($29\times$). Note that both Helmholtz and Klein-Gordon equations contain three second-order derivative terms, while the diffusion equation contains only two. Our results showed the largest differences in runtime and memory usage in Helmholtz and Klein-Gordon equations. This is because SPINN significantly reduces the AD computations with forward-mode AD, implying SPINN is very efficient for solving high-order PDEs.

We investigated the Klein-Gordon experiment further, extending it to a 4-d system by adding one more spatial axis. As shown in Fig. 6d, baseline PINN can only process $23^4$ ($18^4$ for modified MLP) collocation points at once due to a high memory throughput. On the other hand, SPINN can exploit the number of collocation points of $64^4$ ($160\times$ larger than the baseline) to obtain a more precise solution. The exact numerical values are provided in the appendix.

**(2+1)-d Navier-Stokes Equation**    We constructed SPINN to predict the velocity field $u$ and applied forward-mode AD to obtain the vorticity $\omega$. We used the same PDE setting used in causal PINN [43] and compared SPINN against their model. As shown in Tab. 2, our model finds a comparably accurate solution even without the causal loss function. Furthermore, since causal PINN had to iterate over four different tolerance values $\epsilon$ in the causal loss function, they had to run 3~4 times more training epochs, depending on the stopping criterion. As a result, given the same number of collocation points, SPINN converges $60\times$ faster than causal PINN in terms of training runtime. The results of the Navier-Stokes equation ensure that SPINN can successfully solve a chaotic, highly nonlinear PDE. In Fig. 7, we

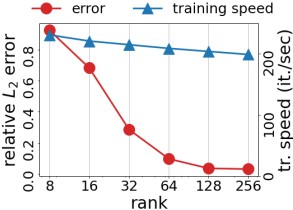

Figure 7: Relative error and training speed of (2+1)-d Navier-Stokes experiment vs. different ranks of SPINN.

Table 2: The numerical result of (2+1)-d Navier-Stokes equation compared against PINN with modified MLP (PINN+mod) and causal PINN. $N_c$ is the number of collocation points. The fourth row shows the training runtime of a single time window in time-marching training. All relative $L_2$ errors are averaged over 3 runs.

† This is the error (the single run result) reported by the causal PINN paper. Since the causal PINN is sensitive to the choice of parameter initialization, we also reported average error obtained from different random seeds by running their official code.

| model | PINN+mod [44] | | causal PINN [43] | | | SPINN (ours) | | |
|---|---|---|---|---|---|---|---|---|
| $N_c$ | $2^{12}$ | $2^{15}$ | $2^{12}$ | $2^{15}$ | $2^{15}$ | $2^{15}$ | $2^{18}$ | $2^{21}$ |
| relative $L_2$ error | 0.0694 ±0.0091 | 0.0581 ±0.0135 | 0.0578 ±0.0117 | 0.0401 ±0.0084 | $0.0353^†$ | 0.0780 ±0.0049 | 0.0363 ±0.0018 | 0.0355 ±0.0006 |
| runtime (hh:mm) | 03:20 | 07:52 | 10:09 | 23:03 | - | 00:07 | 00:09 | 00:14 |
| memory (MB) | 5,198 | 17,046 | 5,200 | 17,132 | - | 764 | 892 | 1,276 |

demonstrate the effect of the rank $r$ for solving the equation. We observed that the performance almost converges at the rank of 128, and increasing the rank does not slow down the training speed too much. Further details and visualization of the predicted vorticity map is shown in appendix section D.4.

**(3+1)-d Navier-Stokes Equation** We proceeded to explore our model on (3+1)-d Navier-Stokes equation by devising a manufactured solution corresponding to the Taylor-Green vortex [41]. It models a decaying vortex, widely used for testing Navier-Stokes numerical solvers [8]. Since the vorticity of the (3+1)-d Navier-Stokes equation is a 3-d vector as opposed to the (2+1)-d system, the equation has three independent components resulting in a total of 33 derivative terms (see section D.5 in the appendix). Similar to the (2+1)-d experiment, the network's output is the velocity field, and we obtained 3-d vorticity by forward-mode AD. Tab. 3 shows the relative error, runtime, and GPU memory. Trained with a single GPU, SPINN achieves a relative error of 1.9e-3 less than 30 minutes. Visualized velocity and vorticity vector fields are provided in the appendix.

Table 3: The numerical result of (3+1)-d Navier-Stokes equation. $N_c$ is the number of collocation points on the entire domain. The relative errors are obtained by comparing the vorticity vector field and averaged over five different runs.

| model | $N_c$ | relative $L_2$ error | runtime (mins) | memory (MB) |
|---|---|---|---|---|
| SPINN | $8^4$ | 0.0090 | 15.33 | 768 |
| | $16^4$ | 0.0041 | 16.83 | 1,192 |
| | $32^4$ | 0.0019 | 26.73 | 2,946 |

## 6 Limitations and Future Works

Although we presented comprehensive experimental results to show the effectiveness of our model, there remain questions on solving more challenging and higher dimensional PDEs. Handling any geometric surface is one of the key advantages of PINNs. In the appendix, we presented a simple way to handle arbitrary domains with SPINN by testing on the Poisson equation in an L-shaped domain. Please see section E.3 for more details. Another method could be applying an additional operation after the SPINN's feature merging to map a rectangular mesh to an arbitrary physical mesh, inspired by PhyGeoNet [11] and Geo-FNO [29]. Combining such techniques for handling arbitrary boundary conditions of complex geometries into ours is an exciting research direction, and we leave it to future work.

We also found that due to its architectural characteristic, SPINN tends to be better trained when the solutions align with the variable separation form (discretized into low-rank tensor). However, note that SPINN is still effective in solving equations where the solution is not a low-rank tensor, such as our examples on diffusion and (2+1)-d Navier-Stokes equation. Please see section D in the appendix for additional experiments and information of which example aligns with the variable separation form or not. Applying SPINN to higher dimensional PDEs, such as the BGK equation, would also have a tremendous practical impact. Lastly, we are still far from the theoretical speed-up (wall-clock runtime vs. FLOPs), which is further discussed in appendix section C. We expect to additionally reduce runtime by optimizing hardware/software, e.g., using customized CUDA kernels.

# 7 Conclusion

We showed a simple yet powerful method to employ large-scale collocation points for PINNs training, leveraging forward-mode AD with the separated MLPs. Experimental results demonstrated that our method significantly reduces both spatial and computational complexity while achieving better accuracy on various three and four-dimensional PDEs. To our knowledge, it is the first attempt to exploit the power of forward-mode AD in PINNs. We believe this work opens up a new direction to rethink the architectural design of neural networks in many scientific applications.

## Acknowledgments and Disclosure of Funding

This research was supported by the Ministry of Science and ICT (MSIT) of Korea, under the National Research Foundation (NRF) grant (2022R1A4A3033571), Institute of Information and Communication Technology Planning Evaluation (IITP) grants for the AI Graduate School program (IITP-2019-0-00421), Korea Institute of Energy Technology Evaluation and Planning (KETEP) and the Ministry of Trade, Industry & Energy (MOTIE) of the Republic of Korea (No. 20224000000360).

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

# Separable PINN - Supplementary Materials

## A  Proof of Theorem 1

Here we show the preliminary lemmas and proofs for Theorem 1. in the main paper. We start by defining a general tensor product between two Hilbert spaces.

**Definition 1.** Let $\{v_\beta\}$ be an orthonormal basis for $\mathcal{H}_2$. **Tensor product** between Hilbert spaces $\mathcal{H}_1$ and $\mathcal{H}_2$, denoted by $\mathcal{H}_1 \otimes \mathcal{H}_2$, is a set of all antilinear mappings $A : \mathcal{H}_2 \to \mathcal{H}_1$ such that $\sum_\beta \|Av_\beta\|^2 < \infty$ for every orthonormal basis for $\mathcal{H}_2$.

Then by Theorem 7.12 in Folland [4], $\mathcal{H}_1 \otimes \mathcal{H}_2$ is also a Hilbert space with respect to norm $\|\cdot\|$ and associated inner product $\langle \cdot, \cdot \rangle$:

$$\|A\|^2 \equiv \sum_\beta \|Av_\beta\|^2, \tag{1}$$

$$\langle A, B \rangle \equiv \sum_\beta \langle Av_\beta, Bv_\beta \rangle, \tag{2}$$

where $A, B \in \mathcal{H}_1 \otimes \mathcal{H}_2$, and $\{v_\beta\}$ is any orthonormal basis of $\mathcal{H}_2$.

**Lemma 1.** Let $x \in \mathcal{H}_1$ and $y, y' \in \mathcal{H}_2$. Then, $(x \otimes y)y' = \langle y, y' \rangle x$.

*Proof.* Let $A$ be a mapping $A : y' \to \langle y, y' \rangle x$, where $\|y\| = 1$. We expand $y'$ with orthonormal basis $\{y, z_\beta\}$. i.e., $\{y, z_\beta\}$ is a basis for $\mathcal{H}_2$. Then,

$$\|Ay\|^2 + \sum_\beta \|Az_\beta\|^2 = \|Ay\|^2 + \sum_\beta \|\langle y, z_\beta \rangle x\|^2 \tag{3}$$

$$= \|Ay\|^2 < \infty \tag{4}$$

It is obvious that $A$ is antilinear. Then by Definition 1, $A \in \mathcal{H}_1 \otimes \mathcal{H}_2$ which is, $Ay' = (x \otimes y)y'$. Therefore, $(x \otimes y)y' = \langle y, y' \rangle x$.

$\square$

**Lemma 2.** $\{u_\alpha \otimes v_\beta\}$ is an orthonormal basis for $\mathcal{H}_1 \otimes \mathcal{H}_2$.

*Proof.* Let $A, B \in \mathcal{H}_1 \otimes \mathcal{H}_2$, where $B = u_\alpha \otimes v_\beta$. Then by the definition of inner product in Eq. 2,

$$\langle A, B \rangle = \sum_i \langle Av_i, Bv_i \rangle \tag{5}$$

$$= \sum_i \langle Av_i, (u_\alpha \otimes v_\beta)v_i \rangle \tag{6}$$

$$= \sum_i \langle Av_i, \langle v_\beta, v_i \rangle u_\alpha \rangle \qquad (\because \text{Lemma 1}) \tag{7}$$

$$= \langle Av_1, \langle v_\beta, v_1 \rangle u_\alpha \rangle + \langle Av_2, \langle v_\beta, v_2 \rangle u_\alpha \rangle + \ldots + \langle Av_\beta, \langle v_\beta, v_\beta \rangle u_\alpha \rangle + \ldots \tag{8}$$

$$= \langle Av_\beta, u_\alpha \rangle \tag{9}$$

Now we check the Parseval identity:

$$\sum_{\alpha, \beta} |\langle A, u_\alpha \otimes v_\beta \rangle|^2 = \sum_{\alpha, \beta} |\langle Av_\beta, u_\alpha \rangle|^2 \qquad (\because u_\alpha \otimes v_\beta = B) \tag{10}$$

$$= \sum_\beta \|Av_\beta\|^2 \tag{11}$$

$$= \|A\|^2. \tag{12}$$

$\therefore \{u_\alpha \otimes v_\beta\}$ is a basis.

$\square$

Now we begin the proof of Theorem 1. in the main paper.

**Theorem 1.** *Let $X, Y$ be compact subsets of $\mathbb{R}^d$. Choose $u \in L^2(X \times Y)$. Then, for arbitrary $\varepsilon > 0$, we can find a sufficiently large $r > 0$ and neural networks $f_j$ and $g_j$ such that*

$$\left\| u - \sum_{j=1}^{r} f_j g_j \right\|_{L^2(X \times Y)} < \varepsilon. \tag{13}$$

*Proof.* Let $\{\phi_i\}$ and $\{\psi_j\}$ be orthonormal basis for $L^2(X)$ and $L^2(Y)$ respectively. Then $\{\phi_i \psi_j\}$ forms an orthonormal basis for $L^2(X \times Y)$ ($\because$ Lemma 2). Therefore, we can find a sufficiently large $r$ such that

$$\left\| u - \sum_{i,j}^{r} a_{ij} \phi_i \psi_j \right\|_{L^2(X \times Y)} < \frac{\varepsilon}{2}, \tag{14}$$

where $a_{ij}$ denotes

$$a_{ij} = \int_{X \times Y} u(x, y) \phi_i(x) \psi_j(y) dx dy.$$

On the other hand, by the universal approximation theorem [2], we can find neural networks $f_j$ and $g_j$ such that

$$\|\phi_i - f_j\|_{L^2(X)} \leq \frac{\varepsilon}{3^j \|u\|_{L^2(X \times Y)}} \quad \text{and} \quad \|\psi_j - g_j\|_{L^2(Y)} \leq \frac{\varepsilon}{3^j \|u\|_{L^2(X \times Y)}}. \tag{15}$$

We first consider the difference between $u$ and $\sum_{i,j}^{r} a_{ij} f_i g_j$:

$$\left\| u - \sum_{i,j}^{r} a_{ij} f_i g_j \right\|_{L^2(X \times Y)} \tag{16}$$

$$\leq \left\| u - \sum_{i,j}^{r} a_{ij} \phi_i \psi_j \right\|_{L^2(X \times Y)} + \left\| \sum_{i,j}^{r} a_{ij} \phi_i \psi_j - \sum_{i,j}^{r} a_{ij} f_i g_j \right\|_{L^2(X \times Y)} \tag{17}$$

$$\equiv I + II \tag{18}$$

Since $|I| < \varepsilon/2$ from (14), it is enough to estimate $II$. For this, we consider

$$\sum_{i,j}^{r} a_{ij} \phi_i \psi_j - \sum_{i,j}^{r} a_{ij} f_i g_j = \sum_{i,j}^{r} a_{ij} \phi_i (\psi_j - g_j) + \sum_{i,j}^{r} a_{ij} (\phi_i - f_i) g_j \tag{19}$$

$$\equiv II_1 + II_2. \tag{20}$$

We first compute $II_1$:

$$\|II_1\|_{L^2(X \times Y)}^2 = \int_{X \times Y} \left\{ \sum_{i,j}^{r} a_{ij} \phi_i (\psi_j - f_j) \right\}^2 dx dy \tag{21}$$

$$= \int_{X \times Y} \left\{ \sum_{j}^{r} \left( \sum_{i}^{r} a_{ij} \phi_i \right) (\psi_j - f_j) \right\}^2 dx dy \tag{22}$$

We set

$$A_j(x) = \sum_{i}^{r} a_{ij} \phi_i(x), \qquad B_j(y) = \psi_j(y) - g_j(y) \tag{23}$$

to write $II_1$ as

$$\|II_1\|_{L^2(X \times Y)}^2 = \int_{X \times Y} \left\{ \sum_j^r A_j(x) B_j(y) \right\}^2 dx dy. \tag{24}$$

We then apply Cauchy-Scharwz inequality to get

$$\|II_1\|_{L^2(X \times Y)}^2 \leq \int_{X \times Y} \left( \sum_j^r |A_j(x)|^2 \right) \left( \sum_j^r |B_j(y)|^2 \right) dx dy \tag{25}$$

$$= \left( \int_X \sum_j^r |A_j(x)|^2 dx \right) \left( \int_Y \sum_j^r |B_j(y)|^2 dy \right). \tag{26}$$

Now

$$\int_X \sum_j^r |A_j(x)|^2 dx = \int_X \sum_j^r \left( \sum_i^r a_{ij} \phi_i \right)^2 dx \tag{27}$$

$$= \sum_j^r \left\| \sum_i^r a_{ij} \phi_i \right\|_{L(X)}^2. \tag{28}$$

Since $\{\phi_i\}$ is an orthonormal basis, we see that

$$\left\| \sum_i^r a_{ij} \phi_i \right\|_{L(X)} \leq \sum_i^r |a_{ij}| \|\phi_i\|_{L(X)} = \sum_i^r |a_{ij}|. \tag{29}$$

Therefore,

$$\int_X \sum_j^r |A_j(x)|^2 dx = \int_X \sum_j^r \left( \sum_i^r a_{ij} \phi_i \right)^2 dx \tag{30}$$

$$= \sum_j^r \int_X \left( \sum_i^r a_{ij} \phi_i \right)^2 dx \tag{31}$$

$$= \sum_j^r \left\| \sum_i^r a_{ij} \phi_i \right\|_{L(X)}^2 \tag{32}$$

$$= \sum_j^r \left( \sum_i^r |a_{ij}| \right)^2. \tag{33}$$

Finally, we recall

$$\left( \sum_i^r |a_{ij}| \right)^2 \leq 2 \sum_i^r |a_{ij}|^2 \tag{34}$$

to conclude

$$\int_X \sum_j^r |A_j(x)|^2 dx \leq 2 \sum_j^r \sum_i^r |a_{ij}|^2 < 2 \sum_{i,j}^\infty |a_{ij}|^2 = 2\|u\|_{L^2(X \times Y)}^2. \tag{35}$$

On the other hand, we have from (15)

$$\int_Y \sum_j^r |B_j(y)|^2 dy = \sum_j^r \int_Y |\psi_j(y) - g_j(y)|^2 dy \tag{36}$$

$$= \sum_j^r \left\| \psi_j(y) - g_j(y) \right\|_{L^2(Y)}^2 \tag{37}$$

$$\leq \sum_j^r \frac{\varepsilon^2}{9^j \|u\|_{L^2(X \times Y)}^2} \tag{38}$$

$$< \frac{\varepsilon^2}{8\|u\|_{L^2(X \times Y)}^2}. \tag{39}$$

Hence we have

$$\|II_1\|_{L^2(X \times Y)}^2 < \frac{\varepsilon^2}{8\|u\|_{L^2(X \times Y)}^2} 2\|u\|_{L^2(X \times Y)}^2 = \frac{\varepsilon^2}{4}. \tag{40}$$

Likewise,

$$\|II_2\|_{L^2(X \times Y)}^2 < \frac{\varepsilon^2}{4}. \tag{41}$$

Therefore,

$$\|II\|_{L^2(X \times Y)}^2 < \frac{\varepsilon^2}{2}. \tag{42}$$

We go back to (16) with the estimates (14) and (42) to derive

$$\left\| u - \sum_{i,j}^r a_{ij} f_i g_j \right\|_{L^2(X \times Y)} < \frac{\varepsilon}{2} + \frac{\varepsilon}{2} = \varepsilon. \tag{43}$$

Finally, we reorder the index to rewrite

$$\sum_{i,j}^r a_{ij} f_i g_j = \sum_i^{\tilde{r}} b_i \tilde{f}_i \tilde{g}_i \tag{44}$$

$$= \sum_i^{\tilde{r}} \left\{ b_i \tilde{f}_i \right\} \tilde{g}_i \tag{45}$$

$$= \sum_i^{\tilde{r}} k_i \tilde{g}_i \tag{46}$$

Without loss of generality, we rewrite $\tilde{r}$, $k_i$, $\tilde{g}_i$ into $r$, $f_i$, $g_i$ respectively, to complete the proof. $\qquad\square$

## B  Training with Physics-Informed Loss

After SPINN predicts an output function with the methods described above, the rest of the training procedure follows the same process used in conventional PINN training [15], except we use forward-mode AD to compute PDE residuals (standard back-propagation, a.k.a. reverse-mode AD for parameter updates). With the slight abuse of notation, our predicted solution function is denoted as $\hat{u}^{(\theta)}(x,t)$ from onwards, explicitly expressing time coordinates. Given an underlying PDE (or ODE), the initial, and the boundary conditions, SPINN is trained with a 'physics-informed' loss function:

$$\min_\theta \mathcal{L}(\hat{u}^{(\theta)}(x,t)) = \min_\theta \lambda_{\text{pde}} \mathcal{L}_{\text{pde}} + \lambda_{\text{ic}} \mathcal{L}_{\text{ic}} + \lambda_{\text{bc}} \mathcal{L}_{\text{bc}}, \tag{47}$$

$$\mathcal{L}_{\text{pde}} = \int_\Gamma \int_\Omega \|\mathcal{N}[\hat{u}^{(\theta)}](x,t)\|^2 dx dt, \tag{48}$$

$$\mathcal{L}_{\text{ic}} = \int_\Omega \|\hat{u}^{(\theta)}(x,0) - u_{\text{ic}}(x)\|^2 dx, \tag{49}$$

$$\mathcal{L}_{\text{bc}} = \int_\Gamma \int_{\partial\Omega} \|\mathcal{B}[\hat{u}^{(\theta)}](x,t) - u_{\text{bc}}(x,t)\|^2 dx dt, \tag{50}$$

where $\Omega$ is an input domain, $\mathcal{N}, \mathcal{B}$ are generic differential operators and $u_{\text{ic}}, u_{\text{bc}}$ are initial, boundary conditions, respectively. $\lambda$ are weighting factors for each loss term. When calculating the PDE loss ($\mathcal{L}_{\text{pde}}$) with Monte-Carlo integral approximation, we sampled collocation points from factorized coordinates and used forward-mode AD. The remaining $\mathcal{L}_{\text{ic}}$ and $\mathcal{L}_{\text{bc}}$ are then computed with initial and boundary coordinates to regress the given conditions. By minimizing the objective loss in Eq. 47, the model output is enforced to satisfy the given equation, the initial, and the boundary conditions.

## C    FLOPs Estimation

The FLOPs for evaluating the derivatives can be systematically calculated by disassembling the computational graph into elementary operations such as additions and multiplications. Given a computational graph of forward pass for computing the primals, AD augments each elementary operation into other elementary operations. The FLOPs in the forward pass can be precisely calculated since it consists of a series of matrix multiplications and additions. We used the method described in [6] to estimate FLOPs for evaluating the derivatives. Table 1 shows the number of additions (ADDS) and multiplications (MULTS) in each evaluation process. Note that FLOPs is a summation of ADDS and MULTS by definition.

One thing to note here is that this is a theoretical estimation. Theoretically, the number of JVP evaluations for computing the gradient with respect to the input coordinates is $Nd$, when $N$ is the number of coordinates for each axis and $d$ is the input dimension (see section 4.3 in the main paper). However, our actual implementation of gradient calculation involves re-computing the feature representations $f^{(\theta_i)}$, which makes the complexity of network propagations from $\mathcal{O}(Nd)$ to $\mathcal{O}(Nd^2)$. Ideally, these feature vectors can be computed only once and stored to be used later for gradient computations. Although it is still significantly more efficient than the conventional PINN's complexity ($Nd^2 \ll N^d$), there is a room to bridge the gap between theoretical FLOPs and actual training runtime by further software optimization.

Table 1: The number of elementary operations for evaluating forward pass, first and second-order derivatives. The calculation is based on $64^3$ collocation points in a 3-d system and the vanilla MLP settings used for diffusion, Helmholtz, and Klein-Gordon equations. We assumed that each derivative is evaluated on every coordinate axis.

| | SPINN (ours) | | PINN (baseline) | |
|---|---|---|---|---|
| | ADDS ($\times 10^6$) | MULTS ($\times 10^6$) | ADDS ($\times 10^6$) | MULTS ($\times 10^6$) |
| forward pass | 20 | 20 | 21,609 | 21,609 |
| 1st-order derivative | 40 | 40 | 86,638 | 43,419 |
| 2nd-order derivative | 80 | 80 | 130,057 | 87,040 |
| MFLOPs (total) | 280 | | 390,370 | |

## D    Experimental Details and Results

In this section, we provide experimental details, numerical results, and visualizations for each experiment in the main paper. Below, we show the list of examples where the solution aligns with the variable separation form or not.
*separable form*: 3-d Helmholtz, (2+1)-d Klein-Gordon, (3+1)-d Klein-Gordon, (3+1)-d Navier-Stokes, (5+1)-d diffusion
*non-separable form*: (2+1)-d diffusion, (2+1)-d Navier-Stokes, (2+1)-d flow mixing, 2-d L-shaped Poisson

### D.1    Diffusion Equation

The diffusion equation is one of the most representative parabolic PDEs, often used for modeling the heat diffusion process. We especially choose a nonlinear diffusion equation where it can be written

as:

$$\partial_t u = \alpha \left( \|\nabla u\|^2 + u\Delta u \right), \qquad\qquad x \in \Omega, t \in \Gamma, \qquad (51)$$

$$u(x,0) = u_{\text{ic}}(x), \qquad\qquad x \in \Omega, \qquad (52)$$

$$u(x,t) = 0, \qquad\qquad x \in \partial\Omega, t \in \Gamma. \qquad (53)$$

We used diffusivity $\alpha = 0.05$, spatial domain $\Omega = [-1,1]^2$, temporal domain $\Gamma = [0,1]$ and used superposition of three Gaussian functions for the initial condition $u_{\text{ic}}$. We obtained the reference solution ($101 \times 101 \times 101$ resolution) through a widely-used PDE solver platform FEniCS [12]. Note that FEniCS is a FEM-based solver. We particularly set the initial condition to be a superposition of three gaussian functions:

$$u_{\text{ic}}(x,y) = 0.25 \exp\left[-10\{(x-0.2)^2 + (y-0.3)^2\}\right] + 0.4 \exp\left[-15\{(x+0.1)^2 + (y+0.5)^2\}\right]$$
$$+ 0.3 \exp\left[-20\{(x+0.5)^2 + y^2\}\right]. \quad (54)$$

For our model, we used three body networks of 4 hidden layers with 64/32 hidden feature/output size each. For the baseline model, we used a single MLP of 5 hidden layers with 128 hidden feature sizes. We used Adam [8] optimizer with a learning rate of 0.001 and trained for 50,000 iterations for every experiment. All weight factors $\lambda$ in the loss function in Eq. 47 are set to 1. The final reported errors are extracted where the total loss was minimum across the entire training iteration. We also resampled the input points every 100 epochs. Tab. 3 shows the numerical results, and the visualized solutions are provided in Fig. 3.

### D.2 Helmholtz Equation

The Helmholtz equation is a time-independent wave equation that takes the form:

$$\Delta u + k^2 u = q, \qquad\qquad x \in \Omega, \qquad (55)$$

$$u(x) = 0, \qquad\qquad x \in \partial\Omega, \qquad (56)$$

where the spatial domain is $\Omega = [-1,1]^3$. For a given source term $q = -(a_1\pi)^2 u - (a_2\pi)^2 u - (a_3\pi)^2 u + k^2 u$, we devised a manufactured solution $u = \sin(a_1\pi x_1)\sin(a_2\pi x_2)\sin(a_3\pi x_3)$, where we take $k = 1, a_1 = 4, a_2 = 4, a_3 = 3$.

For our model, we used three body networks of 4 hidden layers with 64/32 hidden feature/output size each. For the baseline model, we used a single MLP of 5 hidden layers with 128 hidden feature sizes. We used Adam [8] optimizer with a learning rate of 0.001 and trained for 50,000 iterations for every experiment. All weight factors $\lambda$ in the loss function in Eq. 47 are set to 1. The final reported errors are extracted where the total loss was minimum across the entire training iteration. We also resampled the input points every 100 epochs. Tab. 4 shows the numerical results and the visualized solutions are provided in Fig. 4.

### D.3 Klein-Gordon Equation

The Klein-Gordon equation is a nonlinear hyperbolic PDE, which arises in diverse applied physics for modeling relativistic wave propagation. The inhomogeneous Klein-Gordon equation is given by

$$\partial_{tt} u - \Delta u + u^2 = f, \qquad\qquad x \in \Omega, t \in \Gamma, \qquad (57)$$

$$u(x,0) = x_1 + x_2, \qquad\qquad x \in \Omega, \qquad (58)$$

$$u(x,t) = u_{\text{bc}}(x), \qquad\qquad x \in \partial\Omega, t \in \Gamma, \qquad (59)$$

where we chose the spatial/temporal domain to be $\Omega = [-1,1]^2$ and $\Gamma = [0,10]$, respectively. For error measurement, we used a manufactured solution $u = (x_1 + x_2)\cos(2t) + x_1 x_2 \sin(2t)$ and $f$, $u_{bc}$ are extracted from this exact solution.

For our model, we used three body networks of 4 hidden layers with 64/32 hidden feature/output size each. For the baseline model, we used a single MLP of 5 hidden layers with 128 hidden feature sizes. We used Adam [8] optimizer with a learning rate of 0.001 and trained for 50,000 iterations for every experiment. All weight factors $\lambda$ in the loss function in Eq. 47 are set to 1. The final reported errors are extracted where the total loss was minimum across the entire training iteration. We also resampled the input points every 100 epochs. Tab. 5 shows the numerical results.

We used the same settings used in (2+1)-d Klein-Gordon experiment for the (3+1)-d experiment except the manufactured solution was chosen as:

$$u = (x_1 + x_2 + x_3) \cos(t) + x_1 x_2 x_3 \sin(t), \tag{60}$$

where $f$, $u_{bc}$ are extracted from this exact solution. Tab. 6 shows the numerical results. The number of collocation points of $23^4, 18^4$ were the maximum value for PINN and PINN with modified MLP, respectively.

### D.4 (2+1)-d Navier-Stokes Equation

Navier-Stokes equation is a nonlinear time-dependent PDE that describes the motion of a viscous fluid. Various engineering fields rely on this equation, such as modeling the weather, airflow, or ocean currents. The vorticity form for incompressible fluid can be written as below:

$$\partial_t \omega + u \cdot \nabla \omega = \nu \Delta \omega, \qquad\qquad x \in \Omega, t \in \Gamma, \tag{61}$$
$$\nabla \cdot u = 0, \qquad\qquad x \in \Omega, t \in \Gamma, \tag{62}$$
$$\omega(x, 0) = \omega_0(x), \qquad\qquad x \in \Omega, \tag{63}$$

where $u \in \mathbb{R}^2$ is the velocity field, $\omega = \nabla \times u$ is the vorticity, $\omega_0$ is the initial vorticity, and $\nu$ is the viscosity. We used the viscosity 0.01 and made the spatial/temporal domain $\Omega = [0, 2\pi]^2$ and $\Gamma = [0, 1]$, respectively. Note that Eq. 61 models decaying turbulence since there is no forcing term and Eq. 62 is the incompressible fluid condition. The reference solution is generated by JAX-CFD solver [9] which specifically used the pseudo-spectral method. The initial condition was generated using the gaussian random field with a maximum velocity of 5. The resolution of the obtained solution is $100 \times 128 \times 128$ ($N_t \times N_x \times N_y$), and we tested our model on this data.

For our model, we used three body networks (modified MLP) of 3 hidden layers with 128/256 hidden feature/output sizes each. We divided the temporal domain into ten time windows to adopt the time marching method [10, 18]. We used Adam [8] optimizer with a learning rate of 0.002 and each time window is trained for 100,000 iterations. Followed by causal PINN, the PDE (residual) loss and initial condition loss function are written as follows.

$$\mathcal{L}_{\text{pde}} = \frac{\lambda_w}{N_c} \sum^{N_c} |\partial_t w + u_x \partial_x w + u_y \partial_y w - \nu(\partial_{xx} w + \partial_{yy} w)|^2 + \frac{\lambda_c}{N_c} \sum^{N_c} |\partial_x u_x + \partial_y u_y|^2, \quad \text{(64)}$$

$$\mathcal{L}_{\text{ic}} = \frac{\lambda_{ic}}{N_{ic}} \sum^{N_{ic}} \left( |u_x - u_{x0}|^2 + |u_y - u_{y0}|^2 + |w - w_0|^2 \right), \tag{65}$$

where $u_x, u_y$ are $x, y$ components of predicted velocity, $w = \partial_x u_y - \partial_y u_x$, $N_c$ is the number of collocation points, $N_{ic}$ is the number of coordinates for initial condition and $u_{x0}, u_{y0}, u_{w0}$ are the initial conditions. We chose the weighting factors $\lambda_w = 1$, $\lambda_c = 5,000$, and $\lambda_{ic} = 10,000$. We also resampled the input points every 100 epochs. The periodic boundary condition can be explicitly enforced by positional encoding [3], and we specifically used the following encoding function only for the spatial input coordinates.

$$\gamma(x) = [1, \sin(x), \sin(2x), \sin(3x), \sin(4x), \sin(5x), \cos(x), \cos(2x), \cos(3x), \cos(4x), \cos(5x)]^\top. \tag{66}$$

Unlike other experiments, the solution of the Navier-Stokes equation is a 2-dimensional vector-valued function $u : \mathbb{R}^3 \to \mathbb{R}^2$. We can rewrite the feature merging equation Eq. 5 in the main paper to construct SPINN into a 2-dimensional vector function:

$$u_1 = \sum_{j=1}^{r} \prod_{i=1}^{d} f_j^{(\theta_i)}(x_i), \tag{67}$$

$$u_2 = \sum_{j=r+1}^{2r} \prod_{i=1}^{d} f_j^{(\theta_i)}(x_i). \tag{68}$$

For example in our (2+1)-d Navier-Stokes equation setting, $r = 128$ since the network output feature size is 256. This can be applied to any $m$-dimensional vector function if we use a larger output feature

size. More formally, if we want to construct an $m$-dimensional vector function with SPINN of rank $r$, the $k$-th element of the function output can be written as

$$u_k = \sum_{j=(k-1)r+1}^{kr} \prod_{i=1}^{d} f_j^{(\theta_i)}(x_i), \tag{69}$$

where the output feature size of each body network is $mr$.

### D.5 (3+1)-d Navier-Stokes Equation

The vorticity form of (3+1)-d Navier-Stokes equation is given as:

$$\partial_t \omega + (u \cdot \nabla)\omega = (\omega \cdot \nabla)u + \nu \Delta \omega + F, \qquad x \in \Omega, t \in \Gamma, \tag{70}$$
$$\nabla \cdot u = 0, \qquad x \in \Omega, t \in \Gamma, \tag{71}$$
$$\omega(x,0) = \omega_0(x), \qquad x \in \Omega. \tag{72}$$

We constructed the spatial/temporal domain to be $\Omega = [0, 2\pi]^3$ and $\Gamma = [0, 5]$, respectively. We constructed the analytic solution for (3+1)-d Navier-Stokes equation introduced in Taylor et al. [17]. The manufactured velocity and vorticity are

$$u_x = 2e^{-9\nu t} \cos(2x) \sin(2y) \sin(z), \tag{73}$$
$$u_y = -e^{-9\nu t} \sin(2x) \cos(2y) \sin(z), \tag{74}$$
$$u_z = -2e^{-9\nu t} \sin(2x) \sin(2y) \cos(z), \tag{75}$$
$$\omega_x = -3e^{-9\nu t} \sin(2x) \cos(2y) \cos(z), \tag{76}$$
$$\omega_y = 6e^{-9\nu t} \cos(2x) \sin(2y) \cos(z), \tag{77}$$
$$\omega_z = -6e^{-9\nu t} \cos(2x) \cos(2y) \sin(z), \tag{78}$$

where we chose the viscosity to be $\nu = 0.05$. Each forcing term $F$ in the Eq. 70 is then given as

$$F_x = -6e^{-18\nu t} \sin(4y) \sin(2z), \tag{79}$$
$$F_y = -6e^{-18\nu t} \sin(4x) \sin(2z), \tag{80}$$
$$F_z = 6e^{-18\nu t} \sin(4x) \sin(4y). \tag{81}$$

We constructed SPINN to be four body networks (modified MLP) of 5 hidden layers with 64/384 hidden feature/output sizes each. We used Adam [8] optimizer with a learning rate of 0.001 and trained for 50,000 iterations. The weight factors in the loss function in Eq. 48 are chosen as $\lambda_{\text{pde}} = 1$, $\lambda_{\text{ic}} = 10$, and $\lambda_{\text{bc}} = 1$. We also weighted the incompressibility loss (Eq. 71) with 100. The visualized solution vector field is shown in Fig. 6.

## E  Additional Experiments

### E.1 (5+1)-d diffusion Equation

We tested our model on the (5+1)-d diffusion equation to verify the effectiveness of our model on higher dimensional PDE:

$$\frac{\partial u(t, x)}{\partial t} = \Delta u(t, x), \qquad x \in [-1, 1]^5, t \in [0, 1], \tag{82}$$

where the manufactured solution is chosen to be $\|x\|^2 + 10t$. When trained with $8^6$ collocation points, SPINN achieved a relative $L_2$ error of 0.0074 within 2 minutes.

### E.2 (2+1)-d Flow Mixing Problem

This is a time dependent PDE which describes the behavior of two fluids being mixed at the interface in a 2-d environment. Following the settings in CAN-PINNs [1], the equation is

$$\frac{\partial u(t, x, y)}{\partial t} + a\frac{\partial u(t, x, y)}{\partial x} + b\frac{\partial u(t, x, y)}{\partial y} = 0, \qquad t \in [0, 4], x \in [-4, 4], y \in [-4, 4], \tag{83}$$

where

$$a(x,y) = -\frac{v_t}{v_{t,\max}} \frac{y}{r}, \tag{84}$$

$$b(x,y) = \frac{v_t}{v_{t,\max}} \frac{x}{r}, \tag{85}$$

$$v_t = \text{sech}^2(r)\tanh(r), \tag{86}$$

$$r = \sqrt{x^2 + y^2}, \tag{87}$$

$$v_{t,\max} = 0.385. \tag{88}$$

It has an analytic solution [16]:

$$u(t,x,y) = -\tanh\left(\frac{y}{2}\cos(\omega t) - \frac{x}{2}\sin(\omega t)\right), \tag{89}$$

where $\omega = \frac{1}{r}\frac{v_t}{v_{t,\max}}$. The initial and boundary conditions are extracted from the exact solution Eq. 89. When trained with $256^3$ collocation points, SPINN achieved a relative $L_2$ error of 0.0029 in 6 minutes. Figure 1 shows the visualized result.

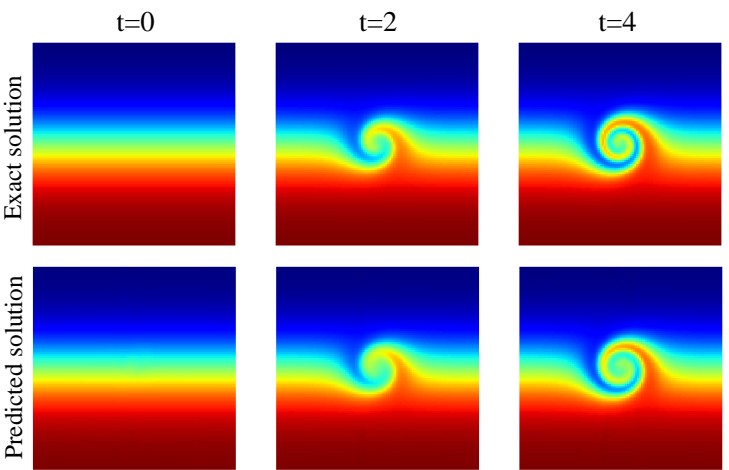

Figure 1: Visualized solution of **flow mixing problem** obtained by SPINN.

### E.3    2-d Poisson equation on an L-shaped domain

Besides the coordinate or mesh transformations [5, 11] mentioned in the limitations section 6, we would like to present yet another simpler way to handle arbitrary domain shapes with SPINN. For boundary points, we train in a non-separable way, just as normal PINN training (non-factorizable coordinates). For collocation points inside the domain, we train in a separable way (factorizable coordinates) and ignore the points outside the domain. This can be easily achieved by finding a tight bounding box of the input domain and masking out the PDE loss when the coordinates are outside.

To show the effectiveness, we tested SPINN on the Poisson equation on an L-shaped domain following the settings used in DeepXDE [13]:

$$-\Delta u(x,y) = 1, \quad (x,y) \in \Omega, \tag{90}$$

$$u(x,y) = 0, \quad (x,y) \in \partial\Omega, \tag{91}$$

where $\Omega = [-1,1]^2 \setminus [0,1]^2$. The reference solution was obtained by the spectral method. SPINN achieved a relative $L_2$ error of 0.0322, while PINNs with modified MLP achieved 0.0392. Figure 2 shows the visualized result. For practical reasons, we split the L-shaped domain into two rectangular domains instead of masking out the top-right part of the domain. The L-shaped domain we showed is just an example case, and we believe that this method can be applied to any arbitrary input domain.

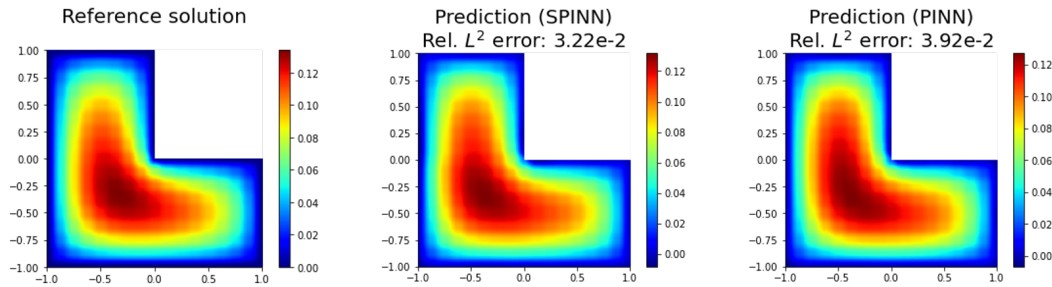

Figure 2: Visualized solution of **Poisson equation** obtained by SPINN and PINNs with modified MLP.

## E.4  Fine Tuning with L-BFGS

We also conducted some experiments to explore the use of L-BFGS when training SPINN. We found that training with Adam first and then fine-tuning with L-BFGS showed a slight increase in accuracy. Note that this training strategy is used by other works [7, 14] and is known to be effective in some cases. Tab. 2 shows the numerical results on three 3-d PDEs. Understanding the effect of the optimization algorithm is still an open question in PINNs, we believe that investigating this issue in the context of SPINN would be a valuable direction for future study.

Table 2: Numerical result of 3-d PDEs with L-BFGS fine-tuning. The number of training collocation points is $64^3$ and a single outer loop L-BFGS is applied.

|  | Diffusion | Helmholtz | Klein-Gordon |
|---|---|---|---|
| Adam | 0.0041 | 0.0360 | 0.0013 |
| Adam + L-BFGS | 0.0041 | 0.0308 | 0.0010 |

Table 3: Full results of **diffusion equation**. $N_c$ is the number of collocation points.

| model | $N_c$ | relative $L_2$ error | RMSE | runtime (ms/iter.) | memory (MB) |
|---|---|---|---|---|---|
| PINN | $16^3$ | 0.0095 | 0.00082 | 3.98 | 1,022 |
|  | $32^3$ | 0.0082 | 0.00071 | 12.82 | 2,942 |
|  | $64^3$ | 0.0081 | 0.00070 | 95.22 | 18,122 |
| PINN + modified MLP | $16^3$ | 0.0048 | 0.00042 | 14.94 | 1,918 |
|  | $32^3$ | 0.0043 | 0.00037 | 29.91 | 4,990 |
|  | $54^3$ | 0.0041 | 0.00036 | 134.64 | 22,248 |
| SPINN | $16^3$ | 0.0447 | 0.00387 | 1.45 | 766 |
|  | $32^3$ | 0.0115 | 0.00100 | 1.76 | 766 |
|  | $64^3$ | 0.0075 | 0.00065 | 1.90 | 766 |
|  | $128^3$ | 0.0061 | 0.00053 | 2.09 | 894 |
|  | $256^3$ | 0.0061 | 0.00053 | 10.54 | 2,174 |
| SPINN + modified MLP | $16^3$ | 0.0390 | 0.00338 | 2.17 | 766 |
|  | $32^3$ | 0.0067 | 0.00058 | 2.44 | 768 |
|  | $64^3$ | 0.0041 | 0.00036 | 2.59 | 768 |
|  | $128^3$ | **0.0036** | 0.00031 | 3.06 | 896 |
|  | $256^3$ | **0.0036** | 0.00031 | 12.13 | 2,176 |

Table 4: Full results of the **Helmholtz equation**. $N_c$ is the number of collocation points.

| model | $N_c$ | relative $L_2$ error | RMSE | runtime (ms/iter.) | memory (MB) |
|---|---|---|---|---|---|
| PINN | $16^3$ | 0.9819 | 0.3420 | 4.84 | 2,810 |
| | $32^3$ | 0.9757 | 0.3398 | 14.84 | 2,938 |
| | $64^3$ | 0.9723 | 0.3386 | 110.23 | 18,118 |
| PINN + modified MLP | $16^3$ | 0.4770 | 0.1661 | 18.32 | 7,034 |
| | $32^3$ | 0.5176 | 0.1803 | 35.02 | 9,082 |
| | $54^3$ | 0.4770 | 0.1661 | 159.90 | 22,244 |
| SPINN | $16^3$ | 0.1177 | 0.0410 | 1.54 | 762 |
| | $32^3$ | 0.0809 | 0.0282 | 1.71 | 762 |
| | $64^3$ | 0.0592 | 0.0206 | 1.85 | 762 |
| | $128^3$ | 0.0449 | 0.0156 | 1.89 | 762 |
| | $256^3$ | 0.0435 | 0.0151 | 3.84 | 1,146 |
| SPINN + modified MLP | $16^3$ | 0.1161 | 0.0404 | 2.24 | 764 |
| | $32^3$ | 0.0595 | 0.0207 | 2.50 | 764 |
| | $64^3$ | 0.0360 | 0.0125 | 2.57 | 764 |
| | $128^3$ | **0.0300** | 0.0104 | 2.76 | 764 |
| | $256^3$ | 0.0311 | 0.0108 | 5.50 | 1,148 |

Table 5: Full results of the **(2+1)-d Klein-Gordon equation**. $N_c$ is the number of collocation points.

| model | $N_c$ | relative $L_2$ error | RMSE | runtime (ms/iter.) | memory (MB) |
|---|---|---|---|---|---|
| PINN | $16^3$ | 0.0343 | 0.0218 | 4.70 | 2,810 |
| | $32^3$ | 0.0281 | 0.0178 | 14.95 | 2,938 |
| | $64^3$ | 0.0299 | 0.0190 | 112.00 | 18,118 |
| PINN + modified MLP | $16^3$ | 0.0158 | 0.0100 | 17.87 | 7,036 |
| | $32^3$ | 0.0185 | 0.0118 | 34.61 | 9,082 |
| | $54^3$ | 0.0163 | 0.0104 | 159.20 | 22,246 |
| SPINN | $16^3$ | 0.0193 | 0.0123 | 1.55 | 762 |
| | $32^3$ | 0.0060 | 0.0038 | 1.71 | 762 |
| | $64^3$ | 0.0045 | 0.0029 | 1.82 | 762 |
| | $128^3$ | 0.0040 | 0.0025 | 1.85 | 890 |
| | $256^3$ | 0.0039 | 0.0025 | 3.98 | 1,658 |
| SPINN + modified MLP | $16^3$ | 0.0062 | 0.0039 | 2.20 | 764 |
| | $32^3$ | 0.0020 | 0.0013 | 2.41 | 764 |
| | $64^3$ | 0.0013 | 0.0008 | 2.57 | 764 |
| | $128^3$ | **0.0008** | 0.0005 | 2.79 | 892 |
| | $256^3$ | 0.0009 | 0.0006 | 5.61 | 1,660 |

Table 6: Full results of the **(3+1)-d Klein-Gordon equation**. $N_c$ is the number of collocation points.

| model | $N_c$ | relative $L_2$ error | RMSE | runtime (ms/iter.) | memory (MB) |
|---|---|---|---|---|---|
| PINN | $16^4$ | 0.0129 | 0.0096 | 43.51 | 5,246 |
| | $23^4$ | 0.0121 | 0.0090 | 154.24 | 22,244 |
| PINN + modified MLP | $16^4$ | 0.0061 | 0.0045 | 100.06 | 17,534 |
| | $18^4$ | 0.0059 | 0.0044 | 174.00 | 22,246 |
| SPINN | $16^4$ | 0.0122 | 0.0091 | 2.45 | 890 |
| | $32^4$ | 0.0095 | 0.0071 | 2.98 | 892 |
| | $64^4$ | 0.0093 | 0.0069 | 9.22 | 2,172 |
| SPINN + modified MLP | $16^4$ | 0.0064 | 0.0048 | 3.48 | 892 |
| | $32^4$ | 0.0022 | 0.0016 | 3.66 | 892 |
| | $64^4$ | **0.0012** | 0.0009 | 10.96 | 2,172 |

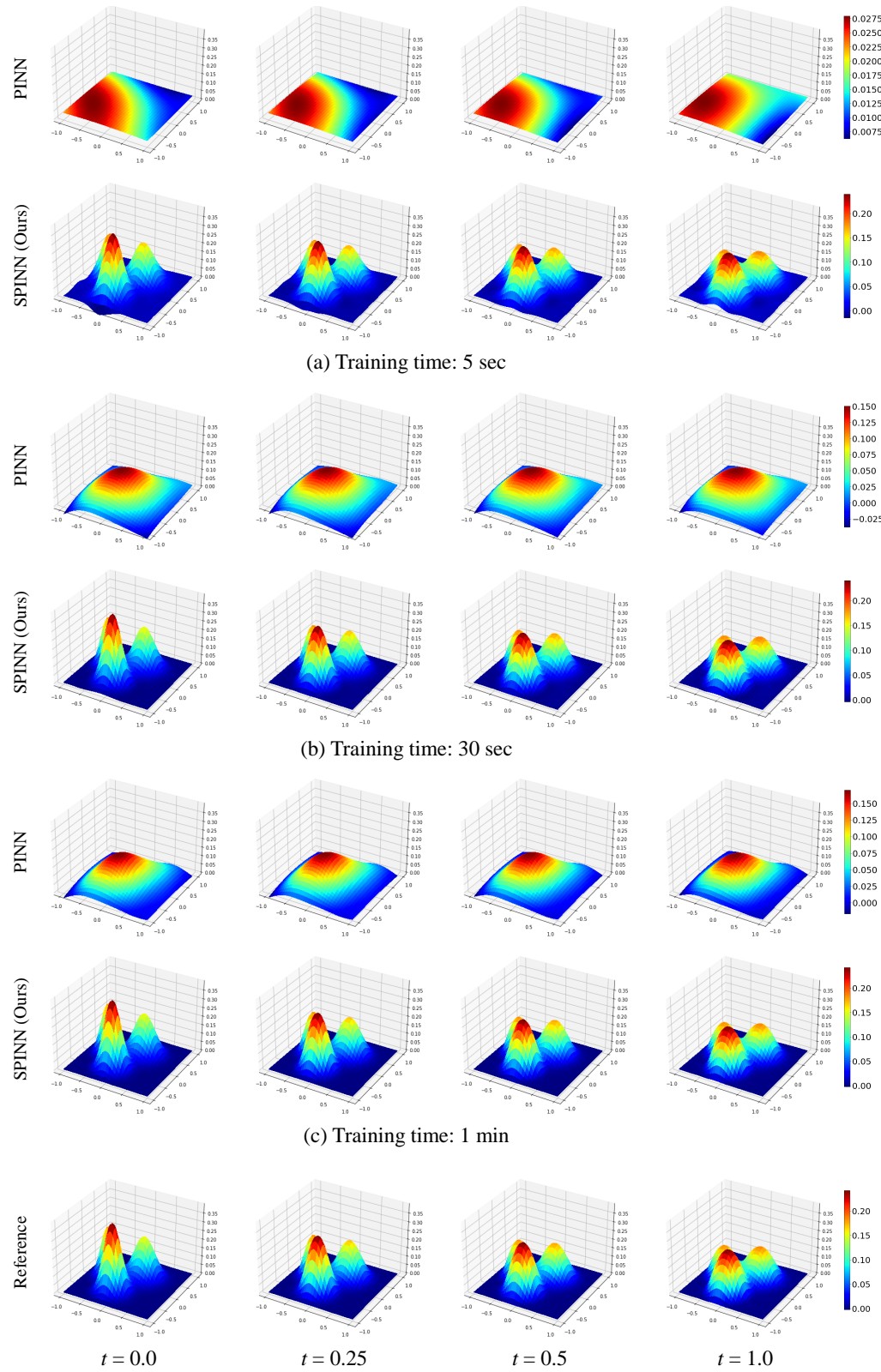

Figure 3: Visualized solution of **nonlinear diffusion equation** obtained by the baseline PINN and SPINN, both trained on $64^3$ collocation points.

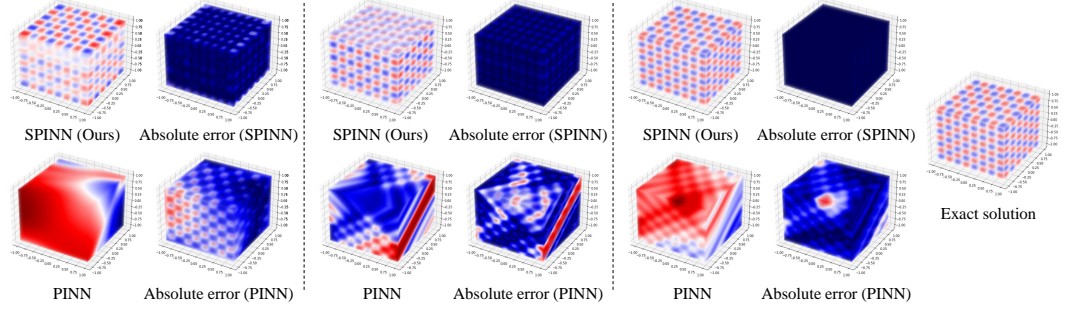

(a) Training time: 5 sec     (b) Training time: 30 sec     (c) Training time: 1 min

Figure 4: Visualized solution of **Helmholtz equation** obtained by the baseline PINN and SPINN, both trained on $64^3$ collocation points.

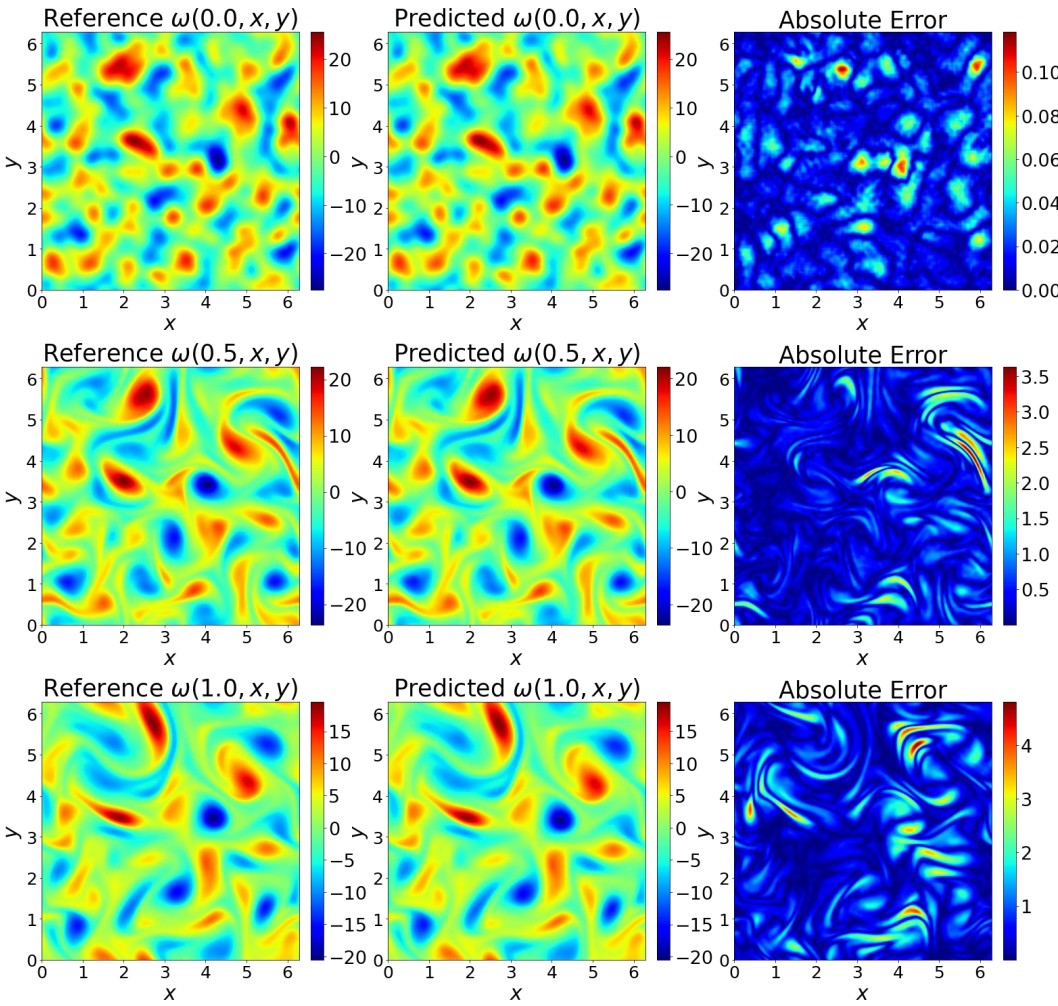

Figure 5: Visualized vorticity maps of **(2+1)-d Navier-Stokes equation** experiment predicted by SPINN. Three snapshots at timestamps $t = 0.0, 0.5, 1.0$ are presented.

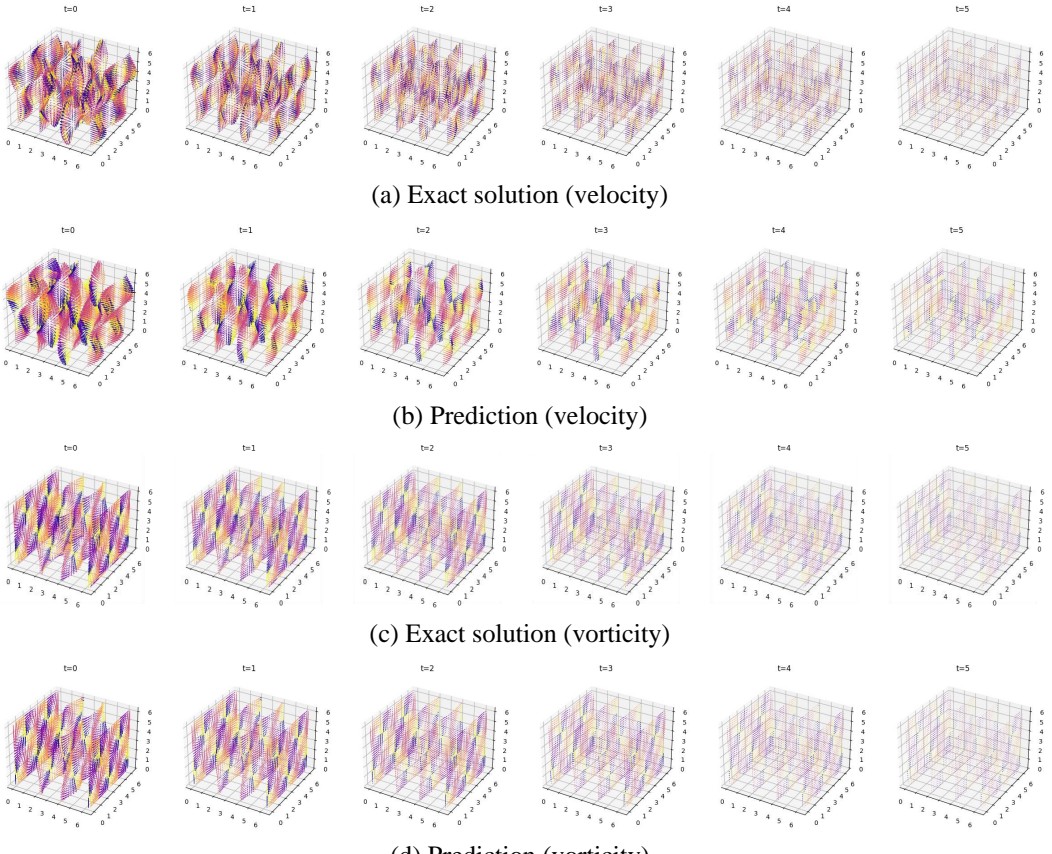

Figure 6: Visualized solution of **(3+1)-d Navier-Stokes equation** obtained by SPINN, trained on $32^4$ collocation points. Each arrow is colored by its zenith angle.

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
