# OpenReview forum: "Separable Physics-Informed Neural Networks"
_NeurIPS.cc/2023/Conference — NeurIPS 2023 spotlight_

### Official Review · Reviewer_dosJ · 2023-07-04

**Soundness:** 2 fair
**Presentation:** 3 good
**Contribution:** 3 good
**Rating:** 5
**Confidence:** 3

**Summary:**

This paper introduces a novel approach to using separable neural networks to represent solutions of partial differential equations (PDEs). In contrast to the conventional form used in physics-informed neural network (PINN) methods, the specialized form of neural networks proposed in this work takes advantage of forward-mode auto-differentiation, leading to faster training times in PINN frameworks. The authors provide a mathematical demonstration of the proposed form's approximation ability and conduct experiments on several PDEs. The results reveal that the proposed method achieves comparable accuracy to previous work, while significantly reducing computational and memory costs.

**Strengths:**

1. The speed-up achieved by the proposed method is highly promising. The time required for training a single network to convergence has been a limitation in the application of PINN methods. This work addresses this issue by introducing a specialized neural network form, and in combination with the forward-mode auto-differentiation method, the training speed is significantly improved.
2. In addition to the improved auto-differentiation computation, this work employs fixed collocation points, enabling further acceleration of the total training time by reusing calculations on each point. By leveraging this approach, the computational efficiency of the method is significantly enhanced.

**Weaknesses:**

Although the proposed method in this paper is novel and interesting, there are several concerns that the reviewer would like to address:

1. The choice of using a manufactured solution that aligns with the variable separation form for evaluating the results on the Helmholtz Equation, Klein-Gordon Equation, and (3+1)-d Navier-Stokes Equation raises some questions. While this choice is suitable for evaluating original PINN methods that do not utilize the variable separation property of the ground truth, it may be unfair to compare SPINN with the original method on these tasks.

2. One of the key advantages of PINN-based methods is their flexibility to be applied to any geometric surface without modifying the grids. However, the variable separation form of the neural network in SPINN may limit this flexibility. It would be challenging for SPINN to fit the boundary conditions of complex geometries due to the constraints of the variable separation form.

3. Throughout the paper, the evaluation metric predominantly used is the $L_2$ evaluation metric. However, the reliability of the $L_2$ metric, especially for non-linear PDEs, can sometimes be questionable. It would be valuable to also report metrics such as the Sobolev norm or the PINN loss to provide a more comprehensive evaluation.

**Questions:**

1. To assess the performance of SPINN on more complex solutions that do not directly align with its form, the authors should conduct more experiments. The experiments could involve using a Multilayer Perceptron (MLP) in the classical PINN as the manufactured solution and training SPINN under the associated PDE.
2. The reviewer is curious about the performance of SPINN on PDEs with sophisticated boundary. For instance, it would be interesting to explore how SPINN performs on a sphere boundary condition, where the variables are still separated into x, y, and z components.
3. Including the Sobolev norm or PINN loss as additional evaluation metrics would provide the reviewer with more information to assess the performance of SPINN comprehensively.
4. The reviewer wonders if it is possible to reduce the dimension of the problem through variable separation. Most PINN methods work on high-dimension manifolds, incurring significant computational costs. While SPINN employs low-dimensional functions to construct the high-dimensional function, its domain is still a high-dimensional manifold. The reviewer believes that there might be a way to directly optimize these low-dimensional functions within the low-dimensional manifold, potentially offering benefits in solving high-dimensional PDEs.

**Limitations:**

The authors have talked their limitation in the paper

---

> ### Author Rebuttal · Authors · 2023-08-09
>
> **R: It is unfair to test SPINN on solutions that align with the variable separation form.**
> A: We tested on manufactured solutions because this is the most conventional way of making an exact reference solution. As you can see in the diffusion and (2+1)-d Navier-Stokes equation, SPINN is still effective in solving such equations where the solution does not align with the variable separation. During the rebuttal period, we conducted additional experiments on the flow mixing problem. This is a (2+1)-d PDE of two fluids being mixed at the interface. We followed the problem settings used in CAN-PINNs [1]. It has an analytical solution that does not explicitly decompose into each input variable:
> $u(t, x, y)=-\tanh(\frac{y}{2}\cos(\omega t)-\frac{x}{2}\sin(\omega t))$, where
> $\omega=\frac{1}{r}\frac{\tanh(r)}{0.385\cosh^2(r)}, r=\sqrt{x^2+y^2}.$
> Since CAN-PINN reported mean-squared errors on a box plot without specifying the exact numerical value, we compared against their rough result. You can see the visualization in Figure 3 of the attached PDF.
> Standard PINNs: 1.08e-2
> CAN-PINN: 2e-5~3e-5
> SPINN: 2.95e-5
> We also added a result of (1+1)-d chaotic Kuramoto-Sivashinsky equation following the causal PINNs’ experiment [2]. Within the temporal domain [0, 0.4], our model achieved a relative L2 error of 3.81e-2 without using a causal loss, and the training speed was 15 times faster than the causal PINNs. You can see the visualized result in our attached PDF, Figure 2. We then have total of 8 PDEs:
> *separable form*: Helmholtz, (2+1)-d Klein-Gordon, (3+1)-d Klein-Gordon, (3+1)-d Navier-Stokes
> *non-separable form*: diffusion, (2+1)-d Navier-Stokes, flow mixing, Kuramoto-Sivashinsky
> Moreover, we will add a paragraph in the experiment section to indicate to the readers that some examples align with the variable separation form or not.
> ***
> **R: SPINN on complex boundary conditions**
> A: Although our current work showed examples of a rectangular domain, it is premature to conclude that SPINN *cannot* be applied to arbitrary geometries. Inspired by PhyGeoNet [3] and Geo-FNO [4], we can apply an additional operation after the SPINN’s feature merging to map a rectangular mesh to an arbitrary physical mesh. We are eager to explore combining these works with SPINN as a future study. We also believe that current SPINN can be applied to circular or spherical boundaries. The exploration of SPINN across a range of coordinate systems greatly intrigues us; however, we could not find a suitable example involving a spherical coordinate-defined PDE with a non-decomposable solution along each axis for testing purposes during the rebuttal phase. We will conduct experiments whenever we find such examples. Thank you for your suggestion.
> ***
> **R: Other metrics**
> A: Based on your and reviewer KaJa’s suggestion, we have introduced two additional metrics – RMSE and PINN loss value – as detailed in the attached PDF file. We will provide the complete results in the revised version of the paper if permitted. Thank you for your valuable comment.
> ***
> **R: Dimensionality reduction through variable separation**
> A: This is a very good idea. We also believe that optimizing the each separated function would greatly reduce the computational cost since it does not require any operations on a high dimensional space. At first thought, if we were able to decompose the original high-dim PDE into a set of ODEs, we could directly optimize each function using the ODE of corresponding dimension. There still exists some challenges that needs to be solved such as 'how can we actually decompose a PDE into ODEs?' or 'can such decomposition applied universally?'. There may be a completely different approach we have not thought of. We find this avenue of research very interesting and are enthusiastic about exchanging further insights with you.
>
> [1] Chiu et al., "CAN-PINN: A fast physics-informed neural network based on coupled-automatic–numerical differentiation method", Computer Methods in Applied Mechanics and Engineering 395 (2022)
> [2] Wang et al., "Respecting causality is all you need for training physics-informed neural networks", arXiv (2022)
> [3] Gao et al., "PhyGeoNet: Physics-Informed Geometry-Adaptive Convolutional Neural Networks for Solving Parameterized Steady-State PDEs on Irregular Domain", Journal of Computational Physics (2021)
> [4] Li et al., "Fourier Neural Operator with Learned Deformations for PDEs on General Geometries", arXiv (2022)

---

> > ### Comment · Reviewer_dosJ · 2023-08-15
> >
> > Thank you for your reply! The reviewer believes that additional evaluation properly addresses weaknesses 1 and 3, but the argument for weakness 2 is not very convincing for the reviewer since those methods are still mesh-based methods. After consideration, the reviewer will increase the score to 5.

---

> > > ### Author Response · Authors · 2023-08-19
> > > **Thank you for your reply**
> > >
> > > Thank you for your reply! We agree that applying the domain transformation on SPINN still has limitations on arbitrary geometry. As you pointed out, the current version of SPINN requires training on a mesh-based grid. We will clarify this in the limitations section. We believe that we can handle this drawback for future work. Thank you.

---

> > > > ### Author Response · Authors · 2023-08-21
> > > > **Handling arbitrary domains**
> > > >
> > > > Besides the coordinate or mesh transformations [1, 2] mentioned in the rebuttal, we would like to present yet another simpler way to handle arbitrary domains with SPINNs. The following is our suggestion.
> > > >
> > > > For boundary points, we train in a non-separable way, just as normal pinn training (non-factorizable coordinates). For collocation points inside the domain, we train in a separable way (factorizable coordinates) and ignore the points outside the domain. We can easily achieve this by finding a tight bounding box of the input domain and masking out the PDE loss when the coordinates are outside.
> > > >
> > > > To show the effectiveness, we trained SPINN on an L-shaped domain and tested on a 2D Poisson equation following the settings used in DeepXDE [3]. The reference solution was obtained by a spectral method (not a manufactured solution), and SPINN achieved a relative L2 error of 0.0322 (PINN got 0.0392). You can see the visualization via an anonymous link we gave to the ACs ('poisson_result.png'). For practical reasons, we split the L-shaped domain into two rectangular domains instead of masking out the top-right part of the domain. Note that the L-shaped domain we showed is just an example case, and we believe that this method can be applied to any arbitrary input domains.
> > > >
> > > > We hope this will resolve the concerns raised by the reviewer, and we are eager to explore further to enable SPINN to handle more complicated domains in future works.
> > > >
> > > > **Reference**
> > > > [1] Gao et al., "PhyGeoNet: Physics-Informed Geometry-Adaptive Convolutional Neural Networks for Solving Parameterized Steady-State PDEs on Irregular Domain", Journal of Computational Physics (2021)
> > > > [2] Li et al., "Fourier Neural Operator with Learned Deformations for PDEs on General Geometries", arXiv (2022)
> > > > [3] Lu et al., "DeepXDE: A deep learning library for solving differential equations", SIAM Review (2021)

---

### Official Review · Reviewer_Zd5Z · 2023-07-04

**Soundness:** 3 good
**Presentation:** 3 good
**Contribution:** 3 good
**Rating:** 6
**Confidence:** 4

**Summary:**

This paper presents a novel variant of PINN for learning multi-dimensional PDEs. Unlike traditional approaches considering point-wise inputs, this method incorporates each axis information separately during the learning process and leverages tensor decomposition to interpret the predicted solution variables. Additionally, the authors consider a forward-mode auto-differentiation technique that facilitates training with a more significant number of collocation points. The extensive experiments proved the efficiency, scalability, and effectiveness of the proposed SPINN.

**Strengths:**

- This paper contributes to alleviating the scalability and efficiency issues of PINNs. Also, the proposed forward-mode auto diff method overcomes collocation point constraints, leading to more accurate solutions. I think it is of interest to the PINN community and the experimental results are convincing.

- The authors provide the theoretical foundation of SPINN. Also, the authors have done extensive experiments on high-dimensional PDEs to prove the scalability of SPINNs. Although the tested PDEs are not that complicated, I don't think that is a big issue in this paper since it is more related to the optimization issue of PINNs.

- This paper is well-written and well-organized.

**Weaknesses:**

- This paper only alleviates the scalability issue, but the primary concern of PINNs lies in the challenges of optimization. Therefore, its contribution to the scientific machine-learning community is moderate.

- It would be good to have a paragraph discussing the related work on scalability for PINNs. There are many existing techniques to handle the scalability issue, such as domain decomposition [1], seq2seq [2], and adaptive sampling method [3]. Using adaptive sampling can also help reduce the collocation points. Moreover, in the baseline comparison, it may strengthen the paper to compare SPINN with one of these techniques though my impression is that SPINN will be better based on the results of this paper.

[1] Jagtap, A. D., & Karniadakis, G. E. (2021, March). Extended Physics-informed Neural Networks (XPINNs): A Generalized Space-Time Domain Decomposition based Deep Learning Framework for Nonlinear Partial Differential Equations. In AAAI spring symposium: MLPS (Vol. 10).

[2] Krishnapriyan, A., Gholami, A., Zhe, S., Kirby, R., & Mahoney, M. W. (2021). Characterizing possible failure modes in physics-informed neural networks. Advances in Neural Information Processing Systems, 34, 26548-26560.

[3] Subramanian, S., Kirby, R. M., Mahoney, M. W., & Gholami, A. (2022). Adaptive self-supervision algorithms for physics-informed neural networks. arXiv preprint arXiv:2207.04084.

**Questions:**

- What about the performance of forward-mode auto diff for higher-order derivatives (e.g., fourth-order in the Kuramoto-Sivashinsky equation)?

- How do the authors consider the rank *r* (i.e., *r*-dimensional feature representation in Eq. (5))? In Theorem 1, the authors claim *r* should be sufficiently large. What about in practice?

- In lines 193-195, for each axis, this paper considers random sampling for 1D input points. Random sampling may not be an optimal choice. Is there any way to improve it in the context of SPINN? Can the authors add some discussions on this part?

- How do the authors consider the optimization in PINNs? In the section on Limitations (lines 335-342), the authors claim that the learning-rate annealing will be considered in the future. I assume the authors were just doing the hyper-parameter tuning by using a grid search. Is that correct?

**Limitations:**

The discussion of the limitations in this paper is comprehensive.

---

> ### Author Rebuttal · Authors · 2023-08-09
>
> **R: The primary concern of PINNs is the optimization issue. Therefore, the contribution is moderate.**
> A: We believe that scalability is also an important issue to tackle in PINNs’ literature. In our Navier-Stokes experiment, where the solution shows complex and turbulent behavior, previous studies [1, 2] have demonstrated the necessity for a large number of collocation points. Particularly in the causal PINNs [1], they had to adopt parallelized training in multi-GPUs and sophisticated software optimization to employ large-scale collocation points. However, without bells and whistles, our proposed method effectively mitigates the scalability issue. Moreover, our experimental results, when compared with prior methods that focused on PINNs' optimization (such as modified MLP, and causal PINNs), suggest that the adoption of large-scale collocation points can serve as an effective means of optimizing PINNs. Standard PINNs were infeasible to employ such large-scale collocation points due to the computational issue. Also, since the seq2seq (time-marching) and the modified MLP are orthogonal to our proposed method, we additionally adopted these techniques to SPINN and observed their positive impact. Other optimization methods like adaptive activation function [3], and learning rate annealing [4] are also applicable to SPINN and we believe that investigating this would be a valuable direction for future study.
> ***
> **R: Related works on scalability for PINNs**
> A: Thank you for the references. We will add an additional paragraph in the related works section. We will also try to run the provided baselines on our PDE settings and compare them with SPINN’s result for future work.
> ***
> **R: Higher-order PDEs**
> A: During the rebuttal period, we tested our model on the chaotic (1+1)-d Kuramoto-Sivashinky equation following the causal PINNs’ experiment [1]. Within the temporal domain [0, 0.4], our model achieved a relative L2 error of 3.81e-2 without using a causal loss, and the training speed was 15 times faster than the causal PINNs. You can see the visualized result in our attached PDF, Figure 2. We also want to notify the reviewer that the PDE we used in (2+1)-d Navier-Stokes equation is a third-order. The vorticity form of the equation is a second-order and our model's prediction is a velocity, in which additional curl operation needs to be taken to obtain the vorticity.
> ***
> **R: In the theorem, rank should be sufficiently large. What about in practice?**
> A: Empirically we showed that SPINN w/ 128 output units (meaning the rank, not too large) can more accurately approximate the (2+1)-d Navier-Stokes equations than any other existing PINNs. As you can see in Figure 7, using a larger rank does not show a meaningful performance increase. Knowing the sufficient number of output units (the rank in SPINN) beforehand is also challenging since we do not know the ranks of the solution functions in general. It is a similar question as 'how many layers and units are sufficient to solve a certain PDE in PINN?'. We believe it is an important question and very challenging, and we plan to further study as a future work.
> ***
> **R: Random sampling may not be an optimal choice. Any other ways to improve in the context of SPINN?**
> A: One straightforward way is to sample more points where the loss value is large. When solving time-dependent PDE, we can sample more temporal coordinates near the initial point (t=0) at the beginning of the training and progressively extend the sampling to later times as the training proceeds. This would guide the model to learn the causal nature of the solution.
> ***
> **R: Hyper-parameter tuning**
> A: Our hyper-parameters are tuned by grid search. We mentioned the learning-rate annealing because it helps the model to search for the best weight balances of the loss terms.
>
> [1] Wang et al., "Respecting causality is all you need for training physics-informed neural networks", arXiv (2022)
> [2] Sankaran et al., "On the impact of larger batch size in the training of Physics Informed Neural Networks", The Symbiosis of Deep Learning and Differential Equations II (2022)
> [3] Jagtap et al., "Adaptive activation functions accelerate convergence in deep and physics-informed neural networks", Journal of Computational Physics (2020)
> [4] Wang et al., "Understanding and mitigating gradient pathologies in physics-informed neural networks", SIAM Journal on Scientific Computing (2021)

---

> > ### Comment · Reviewer_Zd5Z · 2023-08-16
> >
> > Thanks for the clarification and additional experiments. I think the authors have done a good work of tackling the scalability issue of PINNs. I partially agree with the authors that it facilitates optimization with larger amounts of collocation points. I will keep my score.

---

### Official Review · Reviewer_kaJa · 2023-07-05

**Soundness:** 4 excellent
**Presentation:** 4 excellent
**Contribution:** 4 excellent
**Rating:** 8
**Confidence:** 3

**Summary:**

The paper proposes a methodology for significant savings in computation and memory in learning physics informed networks for approximately solving PDEs. This enables learning with a much larger number of points, resulting in better accuracy. The idea is to use a specific function class, where features are constructed from each input dimension, and the output is simply the product of these features. The function class is shown to be universal. Since the same feature encoding can be used for all points that share the same value in each dimension, feature embeddings are calculated only once in a linear time for an exponential number of points, resulting in substantial savings. Using forward mode automatic differentiation, derivatives also benefit from the same savings. Experimental results demonstrate the effectiveness of this approach in approximating the solution of diffusion, Helmholtz, Klein-Gordon, and Navier-Stokes PDEs.

**Strengths:**

The paper is nicely written, the background material is quite useful and many figures help demonstrate the main idea of the paper. In particular, the paper briefly discusses connections to similar ideas in implicit neural representation, and tensor factorization methods, which I found quite relevant.

The main idea is simple, and a priori it is not obvious that the chosen function class can perform so well in practice. Therefore the primary strengths of the paper is its rather surprising experimental results, where for some important PDEs orders of magnitude in speed-up results in accurate estimates compared to vanilla PINN and some other variants. For the same reason, the main theorem of the paper that states the universality of the proposed architecture is also quite useful.

Overall, I enjoyed reading the paper, and I believe the proposed methodology can have a high practical impact.


**Weaknesses:**

While I did not identify a major weakness, I believe the paper can benefit from the following changes:

- While experimental results suggest that the proposed architecture can deliver more improvements in some settings than others (e.g., results for diffusion), there is no discussion of this. In general, while the universality result is reassuring, it is unclear in what kind of PDEs it is a useful inductive bias.

- I found Section 4.3 on gradient computation for SPINN is useful; the paper can benefit from doing the same exercise for higher-order derivatives since the complexity of calculating these derivatives in terms of the dimensionality of the embedding space will dominate the complexity of SPINN.

- While I’m aware that reporting relative error is a common exercise, it is quite sensitive and unstable (due to division by the true value of the PDE), and, therefore, could become misleading. I suggest also reporting the RMSE for all experiments.

- For clarity, the paper should state that the overall complexity still remains exponential in resolution (N), despite reducing it to linear in “network propagations”.


**Questions:**

See above.

**Limitations:**

Yes.

---

> ### Author Rebuttal · Authors · 2023-08-09
>
> **R: Discussion of why SPINN delivers more improvements**
> A: Our performance enhancement mainly stems from the adoption of large-scale collocation points. This is particularly effective when dealing with PDEs where the solution is complex, such as a solution that contains high-frequency components (as demonstrated in our Helmholtz equation) or a solution that shows turbulent and chaotic behaviors (Navier-Stokes equation). Moreover, as the input dimension increases, our model can derive substantial benefits from the adoption of large-scale collocation points, effectively spanning the high-dimensional input space.
> ***
> **R: Higher-order derivatives for section 4.3**
> A: Thank you for your suggestion. We could easily derive the formula for higher-order derivates of Equation 7 and found that as the order increases, the overall pattern in the equation does not change a lot. You can just replace the derivative $df/dx$ with its higher-order counterpart $d^nf/dx^n$. As you mentioned, since computing the derivatives dominates the PINNs training, we will provide a detailed explanation of the higher-order case in this section.
> ***
> **R: Other metrics**
> A: Based on your and reviewer dosJ’s suggestion, we have introduced two additional metrics – RMSE and PINN loss value – as detailed in the attached PDF file. We will provide the complete results in the revised version of the paper if permitted. Thank you for your valuable comment.
> ***
> **R: Clarification in the expressions: “complexity is linear in network propagations”**
> A: Thank you for pointing this out. We will carefully examine the entire text for issues and make the necessary corrections.

---

### Official Review · Reviewer_5Jte · 2023-07-06

**Soundness:** 4 excellent
**Presentation:** 3 good
**Contribution:** 3 good
**Rating:** 8
**Confidence:** 4

**Summary:**

The paper proposes a new class of architecture called Separable PINN (SPINN) that operates on a per-dimension basis instead of the standard coordinate based MLP architecture. This architecture significantly reduces the computational complexity and allows the use of a large number of collocation points to enforce the PDE residuals. The paper also provides an universal approximation theorem for the proposed architecture. The paper empirically demonstrates that the proposed SPINN is significantly faster than the original PINN architecture during training while maintaining accuracy.

**Strengths:**

1. The idea of using a different sub-network for every dimension combined with an outer-product operation to evaluate on a grid is both technically sound and interesting.
2. The paper provides theoretical foundations of the SPINN architecture in the form of the universal approximation theorem.
3. The gradient computation using forward mode AD and the network architecture together leads to significant speed-ups in the training time of PINNs (which is quite impressive).


**Weaknesses:**

1. Although the SPINN architecture is quite efficient, I am concerned about the scalability of the architecture for high dimensional PDEs (>4). (Please see question 1)
2. The size of the plots in Figure 6 is excessively small, making it difficult to compare the performance between the baselines.
3. The PDEs chosen for the baselines are relatively simple (other than the Navier Stokes). It would be interesting to compare the architectures for equations where standard PINNs fail such as Convection Equation (with larger values of $\beta$), Kuramoto Sivashinsky Equation (in chaotic regimes).


**Questions:**

1. One of the main claims of the paper is that the computational complexity of the SPINN architecture linearly scales with the number of collocation points $N$ and the dimensions $d$, in contrast to the standard PINN which scales as $N^d$. However, for the SPINN architecture if we have a different branch for each dimension $d$, the outer product operation of multiplying $d$ different quantities can be numerically unstable. For example, can it lead to gradient vanishing or gradient exploding problems? Is some form of normalization required to ensure that the network is stable?

2. The universal approximation theorem guarantees that the approximation error can be as low as possible. However, does the SPINN architecture have any optimization problems?

3. In Figure 6, for a small number of collocation points(16, 32, 64) for Diffusion Equation, the PINNs perform much better than SPINN. Is there a specific reason why SPINNs did not perform well?


**Limitations:**

The limitations have been adequately addressed in the paper.

---

> ### Author Rebuttal · Authors · 2023-08-09
>
> **R: Computational stability and gradient vanishing/exploding problems**
> A: We have tested on the (5+1)-d system to check whether our model can handle higher dimension PDEs (please see section G.1 in the supplementary material). Additionally, we acknowledge the potential numerical instability introduced by the outer product operation as the dimensionality ($d$) increases. To address this, a promising approach for solving higher-dimensional PDEs involves integrating multiple body networks into one, thereby reducing the multiplications during feature merging:
> $U(x_1, x_2, …, x_d) = \sum\prod f(x_1, x_2)f(x_3, x_4), …, f(x_{d-1}, x_d)$
> We view the exploration of SPINN's performance on high-dimensional PDEs as an intriguing avenue for future research. Thank you for your suggestion.
> ***
> **R: Size of Figure 6**
> A: You can see the magnified version of Figure 6 in our attached PDF file (Figure 1). We will also update the figure in our revised version of the paper. Thank you for your advice.
> ***
> **R: Choice of the PDEs**
> A: The Helmholtz and Klein-Gordon equations pose notable challenges for conventional PINNs, even in 2-dimensional cases [1]. Moreover, we intentionally introduced high-frequency components into the reference solution of the Helmholtz equation to examine the strength of SPINN’s capability to approximate highly complex solutions. Due to neural networks' inherent spectral bias [2], both the standard PINN and the PINN with a modified MLP failed in this case, regardless of the number of collocation points used. During the rebuttal period, we tested our model on the chaotic (1+1)-d Kuramoto-Sivashinky equation following the causal PINNs’ experiment [3]. Within the temporal domain [0, 0.4], our model achieved a relative L2 error of 3.81e-2 without using a causal loss, and the training speed was 15 times faster than the causal PINNs. You can see the visualized result in our attached PDF, Figure 2.
> ***
> **R: Does SPINN architecture has any optimization problems?**
> A: Since our work addresses the scalability issues of PINNs in priority, SPINN inherits the difficulty of optimization in PINNs’ context. However, our experimental results, when compared with prior methods that focused on PINNs optimization (such as modified MLP, and causal PINNs), suggest that the adoption of large-scale collocation points can serve as an effective means of optimizing PINNs. Standard PINNs were infeasible to employ such large-scale collocation points due to the computational issue. Regarding optimization, we have conducted some experiments to explore the use of L-BFGS when training SPINN. Please see section G.2 in the supplementary materials. Understanding the effect of the optimization algorithm is still an open question in PINNs, we believe that investigating this issue in the context of SPINN would be a valuable direction for future study.
> ***
> **R: Why PINNs perform better than SPINN for a small number of collocation points for the diffusion equation?**
> A: Among the PDEs in our experiments, the diffusion equation was the simplest case where the solution is a superposition of three Gaussians. It seems that even standard PINNs trained with a small amount of collocation points can accurately predict the relatively simple solution function. Because of the structural collocation points, SPINN with small training inputs seems to suffer in some cases. However, we want to note that SPINN eventually finds a more accurate solution with more collocation points ($>10^6$). Training standard PINNs in this scale would require much more training time and memory space.
>
>
> [1] Wang et al., "Understanding and mitigating gradient pathologies in physics-informed neural networks", SIAM Journal on Scientific Computing (2021)
> [2] Rahaman et al., "On the spectral bias of neural networks", ICML (2019)
> [3] Wang et al., "Respecting causality is all you need for training physics-informed neural networks", arXiv (2022)

---

> > ### Comment · Reviewer_5Jte · 2023-08-13
> >
> > I thank the authors for providing clarifications to my questions/comments. I have read the other reviews and the authors' comments, and I have increased my score accordingly.

---

### Official Review · Reviewer_3ENg · 2023-07-06

**Soundness:** 3 good
**Presentation:** 3 good
**Contribution:** 2 fair
**Rating:** 5
**Confidence:** 4

**Summary:**

In this paper, the authors propose two main contributions: (i) forward mode AD for PDEs (ii) separating the contributions from each dimension (just like in separable conv) and performing computation using tensor multiplication. The results naturally show significant improvement over PINNs.

**Strengths:**

+ Significant amount of speedup in performance.
+ To be honest, this is probably the only PINN variant I have seen showing results for 3D with as many collocation points as 64^3 etc.
+ They can even solve (3+1) d problems.


**Weaknesses:**

- the language needs to be worked upon. It's not that forward AD is a very new area. In the broad community of Neurips, there are a lot of works that use similar ideas. Statements like "To our knowledge, it is the first attempt to exploit the power of forward-mode AD" needs to be made with caution. Perhaps make it clear that in the purview of PINNs or something. In fact, even in PINN-type papers, there are works that use Finite difference and Finite element based approaches (which don't need a backward pass)
e.g. (i) https://arxiv.org/pdf/2005.08357.pdf (ii) https://arxiv.org/abs/2112.04960 (iii) https://arxiv.org/abs/2211.03241 (iv) https://arxiv.org/abs/1901.06314 (v) https://arxiv.org/pdf/2109.07143.pdf
- lack of rigorous comparisons
- lack of ablation studies

**Questions:**

- What is the rank in Figure 7?
- It seems that your accuracy with even higher number of collocation points is higher than PINNs. Yes, you are solving faster. But is it better? I don't seem to see it clearly...
- Where are the comparisons with other methods like deepOnets, FNOs, etc.
- is 1.9e-3 sufficient loss for Navier-Stokes? What is the typical acceptable error in real applications? and can SPINN achieve it?

---

> ### Author Rebuttal · Authors · 2023-08-09
>
> **R: Expressions when stating our contribution**
> A: In the related work section, we mentioned that we utilized forward-mode AD *in training PINNs* (lines 108-109), but regrettably, this detail was inadvertently omitted in the conclusion section (line 348). Thank you for pointing this out. We will carefully examine the entire text for any additional inaccuracies and make the necessary corrections.
> ***
> **R: Forward-mode AD is not very new**
> A: We want to clarify that our contribution lies in the demonstration of novel PINNs architecture that *utilizes* the forward-mode AD *for fast and accurate PINNs training*. Although forward-mode AD itself is not a new concept, we have rediscovered its potential within the context of PINNs research.
> ***
> **R: PINN-type papers that use numerical derivatives**
> A: We appreciate the provided references. We will cite them in the related works section. It is true that PINN models that use numerical differentiation do not require any backward pass for obtaining the input derivatives. However, these models are still burdened by a computational complexity of $O(N^d)$ during the network propagation, thereby limiting them to handle large-scale collocation points. Furthermore, numerical differentiation has truncation errors depending on the step size.
> ***
> **R: Lack of rigorous comparisons**
> A: We are afraid that we do not know which comparison you would like to see, so we tried our best to give more information in this context. But, please let us know the details of the rigorous comparisons. In the experiment section, we focused on demonstrating 1) training speed and 2) the effectiveness of using large-scale collocation points.
> * Training speed: For a fair comparison, we matched all factors which can affect the training runtime between our model and the baselines. Each experiment was run on a single RTX 3090 GPU and the training source code for both our model and the baselines was implemented with JAX. We matched the total number of learnable parameters, and the training speed of each model is compared under the same number of collocation points.
> * The effectiveness of using large-scale collocation points: We wanted to show that using large-scale collocation points during training can enhance the model’s ability to predict more accurate solutions. We numerically compared the accuracy using the relative L2 error metric. However, as the reviewers ‘KaJa’ and ‘dosJ’ pointed out, we acknowledged the potential limitations of the relative error in universally explaining the results. To address this, we have introduced two additional metrics – RMSE and PINN loss value – as detailed in the attached PDF file. We will provide the complete results in the revised version of the paper if permitted.
> ***
> **R: Lack of rigorous ablation studies**
> A: Again, please let us know which particular ablation study you would like to see. The most essential factor that determines the performance of SPINN is the rank of the reconstructed tensor. We carried out experiments with varying ranks on the (2+1)-dimensional Navier-Stokes equation to showcase its impact on both the prediction accuracy and training speed (Figure 7).
> ***
> **R: Rank in Figure 7**
> A: This is the rank of the reconstructed solution tensor. SPINN constructs multiple rank-1 tensors via the outer product, and the rank is the number of these rank-1 tensors to be added to represent the final solution tensor. Please see Figure 4 in the main paper for details. Increasing the rank gives more expressive power to the model supported by Theorem 1 in the main paper. Figure 7 is intended to demonstrate whether increasing the rank actually helps the model find more accurate solutions.
> ***
> **R: No comparisons against operator methods**
> A: We did not take the neural operator methods into account because they require a pre-training stage to train a neural network over a large set of ground truth data. Thus, the resulting neural network is not capable of solving other PDEs (not shown in the pre-training stage). For example, an FNO trained on Burgers’ equations cannot be used to solve the Helmholtz equations. On the other hand, SPINN (or PINN in general) can be applied to solve any PDEs without any data. To summarize, PINNs and operator learnings (deepOnets, FNO) differ in many aspects, and we thought it is hard to perform an apples-to-apples comparison. We will add more discussions about the differences in the final version of the paper.
> ***
> **R: Typical error in Navier-Stokes equation**
> A: For the (2+1)-d case, we’ve shown that SPINN achieved higher accuracy compared to the causal PINNs, which is the best-performing prior method. We have found a PINNs paper [1] that also tried to solve a (3+1)-d Navier-Stokes equation. Although there is a discrepancy in the equation settings, we leave their result below for a rough reference.
> Equation: (3+1) dimensional Beltrami flow
> Manufactured solution (velocity vectors):
> $u_x = -a[e^{ax}\sin(ay+dz)+e^{az}\cos(ax+dy)]e^{-d^2t}$
> $u_y = -a[e^{ay}\sin(az+dx)+e^{ax}\cos(ay+dz)]e^{-d^2t}$
> $u_z = -a[e^{az}\sin(ax+dy)+e^{ay}\cos(az+dx)]e^{-d^2t}$
> Relative L2 error: 0.023886
> We want to note that the error encountered when solving the (3+1)-d Navier-Stokes equation using a numerical solver can vary significantly based on the specific solver, the chosen discretization methods, the grid resolution, and the nature of the problem to being solved. It is hard to pin down a single “typical” error value as it depends on various factors as mentioned above. Moreover, since there’s no analytical solution available for such a general PDE setting, engineers and researchers often assess the accuracy of their numerical solutions by studying the convergence behavior of the numerical methods, akin to our proposed theorem.
>
> [1] Jin et al., NSFnets (Navier-Stokes flow nets): Physics-informed neural networks for the incompressible Navier-Stokes equations, Journal of Computational Physics (2021)

---

> > ### Comment · Reviewer_3ENg · 2023-08-16
> >
> > With due consideration of the author's response and other reviewers' responses. I would like to increase my score.

---

### Author Rebuttal · Authors · 2023-08-09

## For all reviewers

We thank the reviewers for taking their time to suggest all the invaluable comments and constructive feedback.
**We attached a PDF file** to show tables and visualizations of additional experiments conducted during the rebuttal period.
Please see our responses below.

---

### Author Response · Authors · 2023-08-21
**Handling arbitrary domains**

We thank the reviewers for taking their time to suggest all the invaluable comments and constructive feedback. To address the training of SPINN on arbitrary domains, we provide a description and experimental result obtained during the discussion phase.

Besides the coordinate or mesh transformations [1, 2] mentioned in the rebuttal, we would like to present yet another simpler way to handle arbitrary domains with SPINNs. The following is our suggestion.

For boundary points, we train in a non-separable way, just as normal pinn training (non-factorizable coordinates). For collocation points inside the domain, we train in a separable way (factorizable coordinates) and ignore the points outside the domain. We can easily achieve this by finding a tight bounding box of the input domain and masking out the PDE loss when the coordinates are outside.

To show the effectiveness, we trained SPINN on an L-shaped domain and tested on a 2D Poisson equation following the settings used in DeepXDE [3]. The reference solution was obtained by a spectral method (not a manufactured solution), and SPINN achieved a relative L2 error of 0.0322 (PINN got 0.0392). You can see the visualization via an anonymous link we gave to the ACs ('poisson_result.png'). For practical reasons, we split the L-shaped domain into two rectangular domains instead of masking out the top-right part of the domain. Note that the L-shaped domain we showed is just an example case, and we believe that this method can be applied to any arbitrary input domains.

We hope this will resolve the concerns raised by the reviewer 'dosJ', and we are eager to explore further to enable SPINN to handle more complicated domains in future works.

**Reference**
[1] Gao et al., "PhyGeoNet: Physics-Informed Geometry-Adaptive Convolutional Neural Networks for Solving Parameterized Steady-State PDEs on Irregular Domain", Journal of Computational Physics (2021)
[2] Li et al., "Fourier Neural Operator with Learned Deformations for PDEs on General Geometries", arXiv (2022)
[3] Lu et al., "DeepXDE: A deep learning library for solving differential equations", SIAM Review (2021)

---

### Decision · Program_Chairs · 2023-09-21

**Decision:**

Accept (spotlight)

**Comment:**

In this paper, a new class of architecture of PINN (Physics-informed Neural Network) called Separable PINN (SPINN) is proposed as an alternative to the standard coordinate-based MLP architecture. This significantly reduces the computational complexity of PINNs, enables the use of numerous collocation points, and allows for improved solution accuracy. The proposed architecture also provides a universal approximation theorem. Through experiments, it is demonstrated that SPINN is substantially faster during training than the original PINN while maintaining accuracy. During the rebuttal phase, the authors appropriately addressed reviewers' questions and comments, including additional experiments. While the practicality of PINNs, particularly in high-dimensional problems, is currently limited, this approach achieves remarkable efficiency improvements, making a significant contribution to the advancement of PINNs. This can be regarded as an excellent research achievement.